# The *Ginkgo biloba* L. in China: Current Distribution and Possible Future Habitat

Ying Zhang [1], Jinbing Zhang [1], Li Tian [2,*], Yaohui Huang [3,4] and Changliang Shao [5]

1   College of Geography and Environmental Science, Henan University, Kaifeng 475004, China; zhangying@henu.edu.cn (Y.Z.); 104754200194@henu.edu.cn (J.Z.)
2   National Critical Zone Observatory of Red Soil Hilly Region in Qianyanzhou, Key Laboratory of Ecosystem Network Observation and Modeling, Institute of Geographic Sciences and Natural Resources Research, Chinese Academy of Sciences (CAS), Beijing 100101, China
3   Key Laboratory of Regional Sustainable Development Modeling, Institute of Geographic Sciences and Natural Resources Research, Chinese Academy of Sciences, Beijing 100101, China; huangyaohui0025@igsnrr.ac.cn
4   College of Resources and Environment, University of Chinese Academy of Sciences, Beijing 100049, China
5   Innovation Team of Grassland Ecological Remote Sensing, Institute of Agricultural Resources and Regional Planning of the Chinese Academy of Agricultural Sciences (IARRP-CAAS), Beijing 100081, China; shaochangliang@caas.cn
*   Correspondence: tianli@igsnrr.ac.cn

**Abstract:** With the increase in global temperature, the global change situation dominated by climate warming is becoming more and more serious. Climate change will cause differences in the suitable areas of species in different periods. *Ginkgo biloba* L., a rare and endangered wild plant protected at the national level in China, is the oldest relict plant in the world. Because of severe climate change, only China's wild *Ginkgo biloba* has been preserved, yet China's wild *Ginkgo biloba* population is facing extinction risk. *Ginkgo biloba* has rich ornamental value, application value, economic value, medicinal value and ecological value. Not only can it produce economic and ecological benefits, but it can also produce huge social benefits. Based on the data of *Ginkgo biloba* sample distribution, bioclimatic variables and soil variables, this paper uses the MaxEnt model to simulate *Ginkgo biloba* suitable area under current and future different climate scenarios, and analyzes the changes in the potential suitable area of *Ginkgo biloba* in the future through ArcGIS 10.6. The results are as follows: (1) the results simulated by the MaxEnt model are AUC > 0.9, showing that the simulation results have a high accuracy; (2) the min temperature of the coldest month, precipitation of the wettest month, elevation, and temperature seasonality are the main environmental variables affecting the change in the *Ginkgo biloba* suitable area; (3) under future climate scenarios, the suitable area of *Ginkgo biloba* is predicted to expand in the future, covering most of the south and some northeast regions, and moderate temperature and precipitation changes under climate change are conducive for the growth of *Ginkgo biloba*; and (4) in the future, the distribution center of the suitable area will move to the northeast. According to the conclusions in this paper, it is expected to provide theoretical reference for cultivation and management, sustainable utilization and solution of ecological environment problems of *Ginkgo biloba*.

**Keywords:** *Ginkgo biloba* L.; climate change; MaxEnt; climate change scenario; China

## 1. Introduction

Since the industrial revolution, the rapid development of cities has led to large emissions of greenhouse gases. The IPCC's sixth assessment report (AR6) noted that global temperatures are likely to rise by a maximum of 0.7–4.0 °C relative to 1995–2014 to around 2100 [1]. Climate change is a major challenge [2,3], which will not only have an important impact on the current species distribution, but also have a potential impact on biodiversity and future species distribution [4–6]. Analyzing the species distribution pattern and simulating the future species distribution based on climate change under different scenarios

has become one of the research hotspots of global climate change [7–9], and can provide strategies and a basis for biodiversity conservation and ecosystem development [10].

Climate factors determine the distribution of species at the macro scale. Historical climate changes and regional differences shape the current ecosystem and species distribution patterns [11,12]. In the study of climate change, climate scenario is one of the important studies. The estimated scenario in IPCC scientific assessment reports can provide important scientific basis for government decision-making and various aspects of research [13]. Climate change generally affects species distribution by affecting changes in temperature and precipitation, and different emission scenarios will also lead to higher or lower estimates of the future distribution of the studied species. Since the changes in species distribution caused by climate change are widespread around the world, species distribution models have become an important part of conservation plans to manage biodiversity and ecosystems, and implement prevention strategies effectively and continuously [14].

With the improvement of technology, the special distribution model, based on niche theory, is used in combination with the global climate model to predict species distribution regions in future climate scenarios through species distribution points and environmental variables [15]. The species distribution model is helpful for the appropriate assessment of ecological problems, and its prediction can be used as a basis for conservation planning and decision-making [16,17]. The commonly used species distribution models are BIOCLIM [18], ENFA [19], GARP [20], MaxEnt [21], etc. At present, the MaxEnt model is the optimal solution of species distribution model in terms of accuracy and stability [22]. Its accuracy is higher than other species distribution models, and it is widely used to predict the species potential distribution pattern [23]. It can fit the species distribution area by known sample points and corresponding environmental variables with maximum entropy, and predict the probability of species existence under the influence of environmental variables [21]. Moreover, MaxEnt is suitable for some cases where the data information is not clear [24], and is therefore suitable for simulating species distribution [25,26]. Gebrewahid et al. [27] applied MaxEnt to accurately predict *Oxytenanthera abyssinica* (A. Richard) distribution under future climate change; Zhang et al. [28] obtained the global potential risk area of the alien species *Xanthium italicum* through the MaxEnt model, which is helpful for countries to prevent and control; Li et al. [29] used the MaxEnt model to predict the current and future distribution of three *Coptis* herbs (*Coptis chinensis* Franch., *Coptis deltoidei* C. Y. Cheng et Hsiao and *Coptis teeta* Wall) in China under climate change. At present, MaxEnt has been applied to the prevention of alien species invasion [30], the restoration and protection of endangered species [31], the simulation and prediction of species distribution [23], the planning of protected areas [10], etc.

*Ginkgo biloba* L. *(Ginkgo biloba)* has a high requirement for hydrothermal conditions during its growth and is suitable for growing in places with sufficient light. Ginkgo is an important medicinal tree species. Extracts from its leaves have a good effect on the treatment of neurodegenerative diseases, cardiovascular diseases, cancer, stress, memory loss, tinnitus and psychosis (such as schizophrenia) [32,33], and are the focus of attention in the field of herbal medicine. In addition to its medicinal value, *Ginkgo biloba* has many economic values such as food, ecology, environmental protection, and natural landscape [34]. It has indispensable value in people's production and life. Most scholars' research on the *Ginkgo* tree mainly focused on the fields of pharmacological agents, biochemistry, and molecular biology. There are few studies on the effects of climate change on *Ginkgo biloba*; previous studies have been conducted under climate models previously evaluated by the IPCC. Compared with the latest climate assessment model, the effect is poor. In addition, although Ginkgo is common, it is actually an ancient relict plant in the world [35]. It is the only surviving organism of the genus *Ginkgo* in the family *Ginkgoaceae* and is susceptible to dramatic climate changes. More than half a million years ago, *Ginkgo biloba* was extinct in most parts of Europe, North America, and Asia due to climate disturbance. Only China's preservation continues to this day, while most of the existing *Ginkgo biloba* in other countries

is directly or indirectly introduced from China [36]. Due to the uniqueness of *Ginkgo biloba*, it was included in the 'IUCN Red List of Endangered Species' in 1998 [37].

Overall, based on the MaxEnt model, there is a lack of research on wild *Ginkgo biloba* distribution changes in China under global warming. Therefore, based on the MaxEnt model, this paper selects four different concentrations of greenhouse gas emission scenarios in the middle and late 21st century (2050s and 2070s) and predicts *Ginkgo biloba* geographical distribution pattern in future climate change (Figure 1) to provide the theoretical basis for the protection and management of *Ginkgo biloba*, sustainable growth and development. and ecological environment problems, as well as the protection of endangered species.

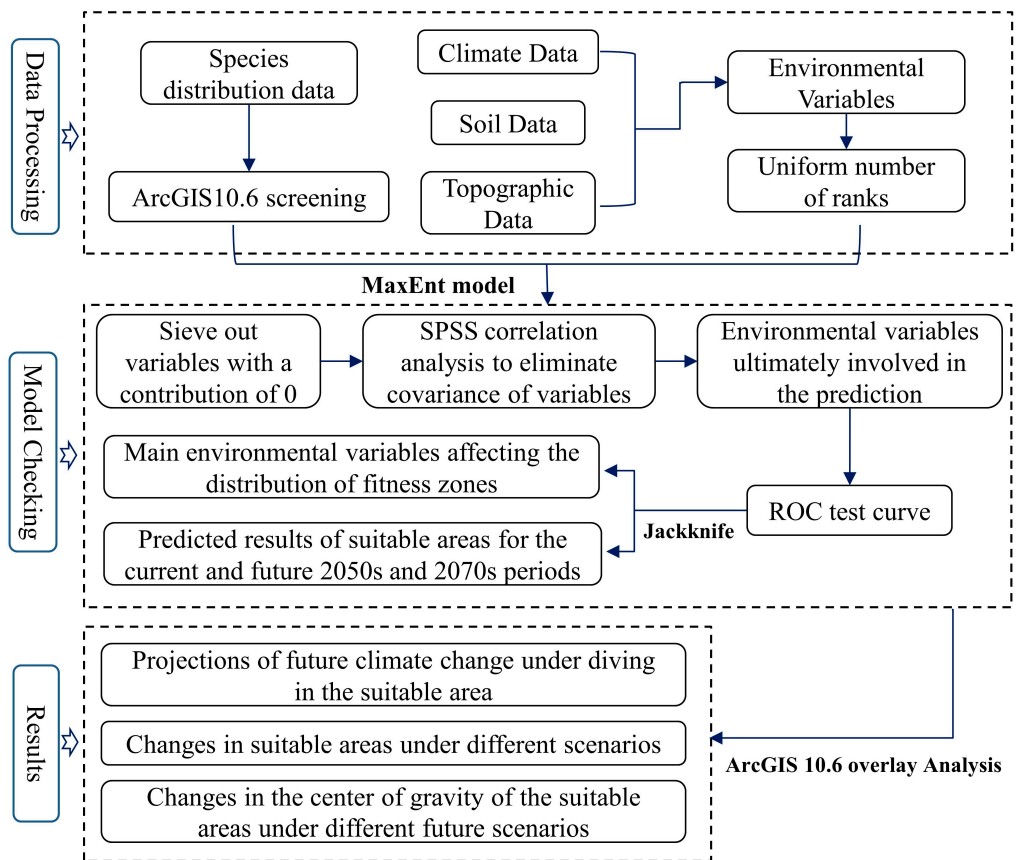

**Figure 1.** Research framework.

## 2. Materials and Methods

### 2.1. Data Sources

#### 2.1.1. *Ginkgo biloba* Sample Distribution Data

The distribution data of *Ginkgo biloba* used in this paper are mainly derived from the GBIF database (http://www.gbif.org/, accessed on 17 May 2022) (167 points) and the NSII (http://mnh.scu.edu.cn/, accessed on 17 May 2022) (124 points), and the recorded data of *Ginkgo biloba* from 1970 to 2020 were obtained. After processing, the sample points with repeated coordinates and the sample points with wrong geographical distribution were screened out. Then, due to the 1 km resolution of the variable data involved in the analysis, we retained only one of the two sample points with a ground distance of less than 2 km by buffer analysis. Finally, 209 effective sample points were retained for simulation analysis (Figure 2).

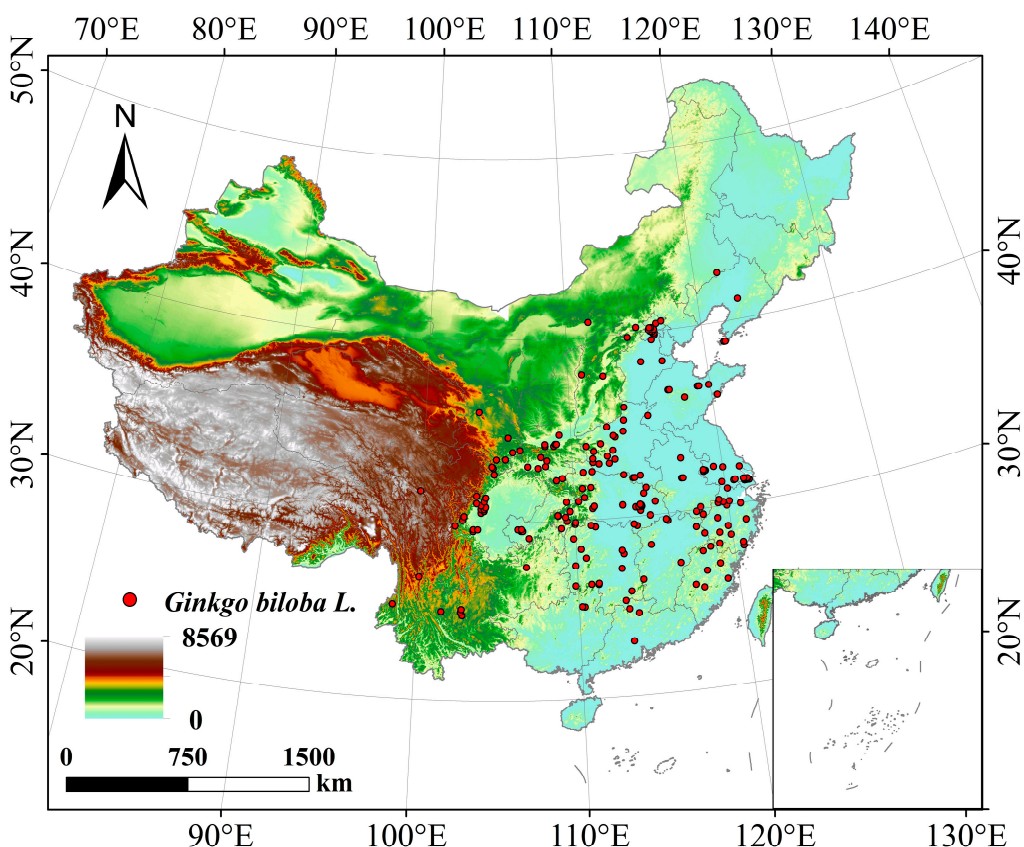

**Figure 2.** *Ginkgo biloba* L. sample distribution map.

2.1.2. Environmental Variables Data

Plant growth is closely related to climate, topography, soil properties, vegetation types, NDVI and so on. In this study, the current and future distribution patterns of *Ginkgo biloba* will be simulated. Considering the availability of data in these periods (since NDVI data in the future period cannot be obtained), we selected three environmental variables: bioclimate, soil and topography [38]. The bioclimatic variables data (1970–2000, 2041–2060/2050s period, 2061–2080/2070s period) in this paper are derived from the WorldClim 2.1 (http://www.worldclim.org/, accessed on 3 June 2022) CMIP6 model data. Compared with the CMIP5 version of the climate model, the simulation ability of the BCC-CSM2-MR for China's climate has been greatly improved [39]. Therefore, the future bioclimatic variable factors were selected under the BCC-CSM2-MR module, four different concentrations of greenhouse gas emission scenarios (ssp126 is a low-forcing scenario, ssp245 is a medium forcing scenario, ssp370 is a medium-to-high-forcing scenario, and ssp585 is a high-forcing scenario [40,41]) in the 2050s period (2041–2060 average) and the 2070s period (2061–2080 average) were used. These two future periods are relatively suitable for the time span from the current period. It is convenient to take corresponding biological protection measures in time according to the prediction results. These global climate variable data include 19 bioclimatic variables, see File S1 for specific data.

In addition to climatic factors, topographic also has a significant effect on plants. Temperature, oxygen content, air pressure and so on are closely related to altitude. The slope direction reflects the solar exposure conditions of the plant growing environment. The topographic variables used in this paper are from the NCDC (http://www.ncdc.ac.cn/, accessed on 3 June 2022) [42].

Secondly, plant growth is inseparable from the static variable of soil [38]. In this paper, 36 soil environmental data [43] from the NCDC (http://www.ncdc.ac.cn/, accessed on 7 June 2022) are also selected. This data is derived from the HWSD.

Finally, China's vector data (GS (2019)1822) comes from the RESDC (http://www.resdc.cn/, accessed on 3 June 2022). The research in this paper assumes that soil variables and topographic variables do not change during the simulation of potential geographical distribution. Resolution of the above raster data is 30 s.

## 2.2. Methods

In this paper, the MaxEnt3.4.4 (Maximum Entropy Model, https://biodiversityinformatics.amnh.org/open_source/maxent/, accessed on 17 December 2022) is selected for simulation analysis. The MaxEnt model is a mathematical method for inferring unknown probability distribution based on limited known information. It has been widely used as a quantitative analysis tool for predicting the future distribution of species.

### 2.2.1. Evaluate the Main Environmental Variables

To prevent the multicollinearity of environmental variables in the process of model prediction and leading to over-fitting the model results [44], variables should be screened first. Therefore, the following measures were taken to complete the screening of environmental variables. Firstly, 58 environmental variables were operated by default in MaxEnt, removing the environmental variables with a contribution of 0 (S_ECE, S_PH_H2O, T_CASO4, T_ECE, T_REF_BULK). Secondly, for the remaining environment variables, Spearman correlation analysis was performed in IBM SPSS Statistics 26.0. Then, environmental variables with a correlation ≤ 0.8 were selected. For the correlation above 0.8, the large contribution was retained [45], excluding variables with a small contribution rate. Finally, 32 environmental variables are obtained for research and analysis (Table 1).

**Table 1.** Contribution rate and permutation important value of environmental variables used to predict the potential geographical distribution of *Ginkgo biloba*.

| Data Type | Environment Variable | Contribution Rate (%) | Permutation Importance |
|---|---|---|---|
| Climate variables | Bio2 | 1.4 | 0.7 |
| | Bio3 | 2 | 1.4 |
| | Bio4 | 3 | 0.8 |
| | Bio6 | 43.5 | 52.1 |
| | Bio10 | 0.3 | 0.4 |
| | Bio13 | 21.8 | 9.7 |
| | Bio14 | 0.6 | 2.4 |
| | Bio15 | 2.7 | 2.2 |
| Soil variables | AWC_CLASS | 1.8 | 1.4 |
| | DRAINAGE | 2.7 | 1 |
| | REF_DEPTH | 0.2 | 0 |
| | S_CACO3 | 1.3 | 1.3 |
| | S_CASO4 | 0.1 | 0.1 |
| | S_CEC_SOLT | 0.5 | 0.7 |
| | S_CLAY | 0.2 | 1 |
| | S_ESP | 0.7 | 1.4 |
| | S_GRAVEL | 0.7 | 0.7 |
| | S_OC | 0.1 | 0.1 |
| | S_SAND | 0.2 | 0.9 |
| | S_USDA_TEX | 1.5 | 0.8 |
| | T_CEC_CLAY | 0.4 | 0.7 |
| | T_CEC_SOLT | 0.2 | 0.3 |
| | T_CLAY | 0.3 | 1.2 |
| | T_ESP | 0.3 | 0.1 |
| | T_GRAVEL | 1.9 | 2.7 |
| | T_OC | 0.1 | 0.2 |
| | T_SILT | 0.6 | 1.3 |
| | T_TEXTURE | 0.3 | 0 |
| | T_USDA_TEX | 1.1 | 0.6 |
| Terrible variables | DEM | 4.6 | 8.9 |
| | Slop | 2.9 | 3.3 |
| | Aspect | 2.2 | 1.3 |

### 2.2.2. Model Accuracy Validation

The processed *Ginkgo biloba* sample distribution data and 32 environmental variables were imported into the MaxEnt, and the Random seed was selected. Through the setting evaluation of regularization multiplier value and feature class, we set the minimum combination of AICc (Akaike Information Criterion) as the best model [46]. The Bootstrap replicates were set to 15, and 70% of the sample data was randomly selected as a training set for model establishment. The remaining 30% were used as the test set [47]; the maximum number of background points was 10,000 [10,29,47]. In the past, most scholars chose to set the number of repeated trainings to 10 times [48,49], but in this paper, after several pre-simulations in the MaxEnt model, it was found that the output potential distribution results of *Ginkgo biloba* are more random, so according to the number of occurrences, the cross-validation method was used for model evaluation, and the number of repeated training was set to 15 so that the final output results were most likely balanced and reliable.

The importance of environmental variables to *Ginkgo biloba* distribution can be evaluated by Jackknife, percentage contribution rate and permutation importance value. The ROC evaluates the accuracy of the model with the area under the curve (AUC). ROC analysis method has been widely used in the evaluation of species distribution prediction models [27,29,31,50,51]. An AUC value of 0.5–0.6 is predicted failure; 0.6–0.7 indicates poor prediction; 0.7–0.8 represents a general forecast; 0.8–0.9 indicates good prediction; and 0.9–1.0 represents that the prediction results are very accurate and reliable [52]. When AUC > 0.85, the predicted results are acceptable [53,54].

### 2.2.3. Division of Suitable Area

Through ArcGIS 10.6, we transformed the simulation results of the model into raster layers to obtain the existence probability map (0–1) of *Ginkgo biloba*. The closer the existence probability is to 1, the better the condition of species survival. Then, based on Maximum test sensitivity plus specificity area, the *Ginkgo biloba* suitable area was divided [55]: <0.23 is an unsuitable area, 0.23–0.4 is a less suitable area, 0.4–0.6 is a moderately suitable area, and >0.6 is a highly suitable area.

To facilitate the clear and intuitive observation of the changes of the *Ginkgo biloba* suitable area, the probability distribution map of *Ginkgo biloba* species (TIF format) was divided into different suitable areas according to the grade. Then, the distribution map of the suitable area in different periods in the future can be superimposed with the distribution map of the current period, and the grid calculator tool can be used to obtain the spatial changes of *Ginkgo biloba* suitable areas.

### 3. Results

*3.1. Accuracy Evaluation of Model Prediction*

The results of AUC obtained from MaxEnt (Table 2) indicate that MaxEnt has good prediction accuracy and credibility [56], and can be used to simulate *Ginkgo biloba* distribution.

**Table 2.** The AUC values of simulation results.

|  | AUC of Training Date | AUC of Test Date | AUC of Random Prediction |
|---|---|---|---|
| Current | 0.9402 | 0.9050 | 0.5 |
| 2050s_ssp126 | 0.9421 | 0.9040 | 0.5 |
| 2050s_ssp245 | 0.9411 | 0.9124 | 0.5 |
| 2050s_ssp370 | 0.9390 | 0.9090 | 0.5 |
| 2050s_ssp585 | 0.9421 | 0.9080 | 0.5 |
| 2070s_ssp126 | 0.9425 | 0.9040 | 0.5 |
| 2070s_ssp245 | 0.9446 | 0.9120 | 0.5 |
| 2070s_ssp370 | 0.9404 | 0.9060 | 0.5 |
| 2070s_ssp585 | 0.9383 | 0.9001 | 0.5 |

### 3.2. Main Environmental Variables Affecting the Ginkgo biloba Suitable Area

The Jackknife test can determine the predictive ability of each variable [57,58] (Figure 3). From Figure 3, the environmental variables with a larger regularized training gain are Bio6 (0.99), Bio2 (0.72), Bio13 (0.62), Bio10 (0.54), Bio14(0.50), and DEM (0.48). The environmental variables with a larger regularized test gain are Bio6 (1.06), Bio2 (0.78), Bio13 (0.70), Bio10 (0.57), Bio14 (0.56), DEM (0.50), and Bio4 (0.41). The environmental variables with a larger AUC value are Bio6 (0.87), Bio2 (0.82), Bio13 (0.81), Bio14 (0.78), DEM (0.77), Bio15 (0.74), Bio4 (0.74), T_USDA_TEX (0.72), and Bio3 (0.71).

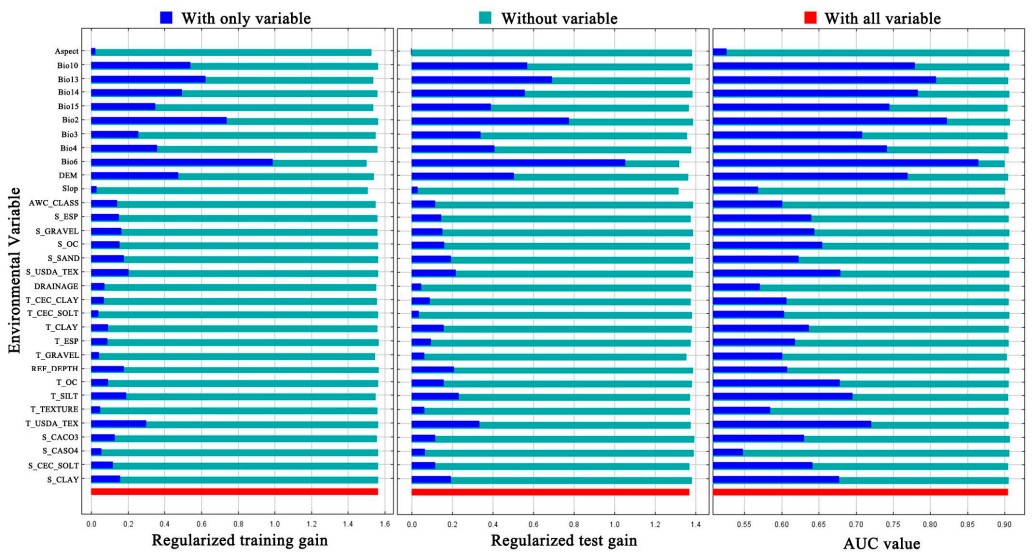

**Figure 3.** The result of Jackknife test.

The output results of the model (Table 1) showed that the contribution of each environmental variable ≥ 3% are Bio6 (min temperature of the coldest month, 43.5%), Bio13 (precipitation of the wettest month, 21.8%), DEM (4.6%), and Bio4 (temperature seasonality, 3%), with a cumulative contribution of 72.9%. The permutation importance values of ≥3% for each environmental variable are Bio6 (52.1%), Bio13 (9.7%), DEM (8.9%), and Slop (3.3%), with a cumulative permutation importance value of 74%. Combining the analysis of Jackknife, percentage contribution and replacement importance values, the dominant environmental variables affecting *Ginkgo biloba* are Bio6, Bio13, DEM, and Bio4. According to the output results of MaxEnt (Figure 4), using the presence probability of 0.23 as the critical condition for the *Ginkgo biloba* suitable area, the threshold values of the dominant environmental variables are: Bio6, −12.86–6.60 °C; Bio13, 120.68–544.31 mm; DEM, 0–1971.40 m; Bio4, 137.81–1219.23. According to the classification of the hierarchy of the probability of existence of the *Ginkgo biloba* suitable area, the dominant environmental variable thresholds affecting the *Ginkgo biloba* highly suitable area are: Bio6,−7.21–3.18 °C; Bio13, 162.04–307.97 mm; DEM, 5.9–501.70 m; Bio4, 691.35–977.29.

### 3.3. Simulation of Distribution of Ginkgo biloba Suitable Area under Climate Change Scenarios

3.3.1. Distribution of Suitable Area for *Ginkgo biloba* under Current Climate Scenario

Based on the results of the *Ginkgo biloba* suitable area distribution output from the MaxEnt model, the current distribution of the *Ginkgo biloba* suitable area is obtained by the reclassification tool of ArcGIS 10.6 (Figure 5), and the suitable area of *Ginkgo biloba* is distributed in Beijing, Hebei, Shandong, Henan, Chongqing, Hubei, Anhui, Jiangsu, Shanghai, Zhejiang, Jiangxi, Hunan, and most of Shaanxi, Sichuan, Guizhou, and small parts of Gansu, Liaoning, Fujian, Guangdong, and Guangxi in China. Among them, the highly suitable area is distributed in Hunan, Hubei, Anhui, Jiangxi, and Henan, and mainly scattered in Shaanxi, Sichuan, Jiangsu and Hebei. In general, currently, the suitable area of

China's *Ginkgo biloba* is mainly 24°–41° N, 102°–124° E, spreading over most of the central region and a small part of the southern and northern high-altitude regions.

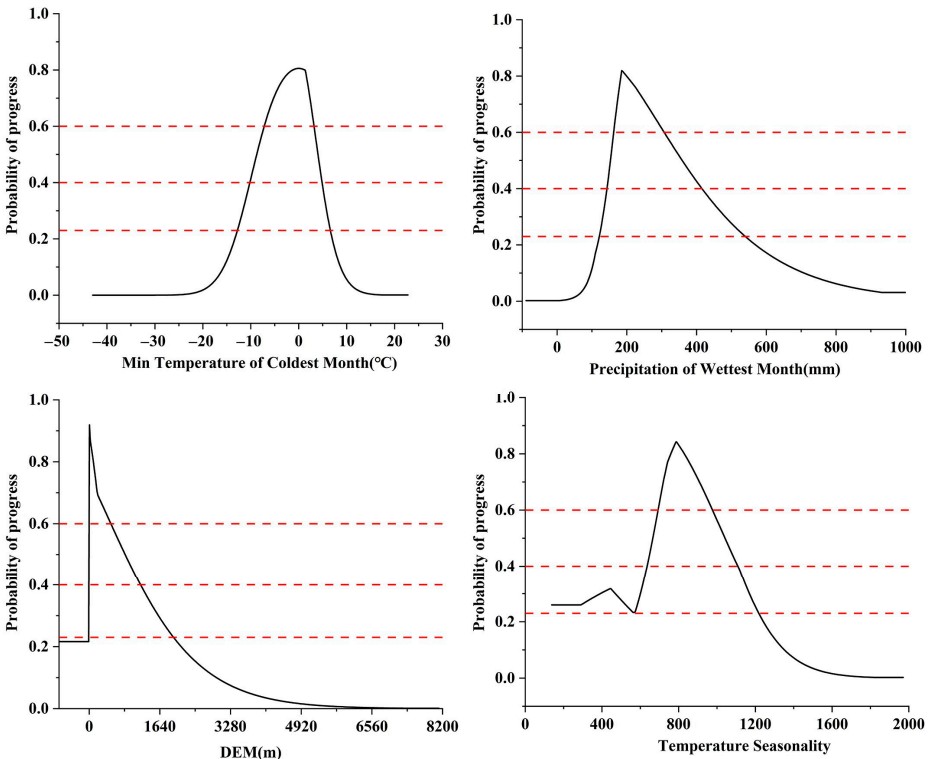

**Figure 4.** Response curves of dominant environmental variables (The red dot line indicates the division of the probability of the existence of *Ginkgo biloba* suitable areas, from bottom to top: 0.23, 0.4, 0.6).

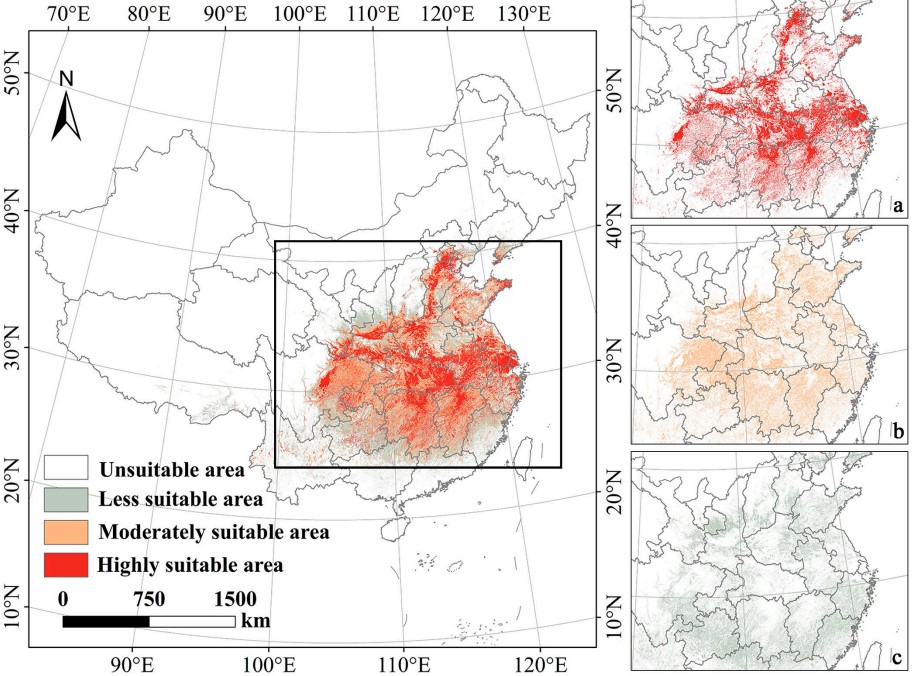

**Figure 5.** The suitable distribution area of *Ginkgo biloba* under the current climate scenario ((**a**) a highly suitable area; (**b**) a moderately suitable area; (**c**) less suitable areas).

Through ArcGIS10.6, the area of different suitable areas under each scenario is extracted (Table 3). From Table 3, the total area of the *Ginkgo biloba* suitable area is currently $214.19 \times 10^4$ km$^2$, and the less suitable area is $80.81 \times 10^4$ km$^2$, accounting for 37.73%, while the moderately suitable area and highly suitable area are $69.02 \times 10^4$ km$^2$ and $64.36 \times 10^4$ km$^2$. The two areas are equivalent, accounting for 32.22% and 30.05%. Among them, the largest area of suitable area is Hunan Province, with $20.76 \times 10^4$ km$^2$; the largest area of highly suitable area is Hubei Province, with $9.73 \times 10^4$ km$^2$; the largest area of moderately suitable area is Sichuan Province, with $8.07 \times 10^4$ km$^2$; the largest area of less suitable area is Guizhou Province, with $8.28 \times 10^4$ km$^2$, accounting for 10.24% of the total area of less suitable area. The largest area of the less suitable area is Guizhou Province, with $8.28 \times 10^4$ km$^2$. The simulation of the *Ginkgo biloba* geographic distribution pattern under current climate conditions is basically consistent with the distribution of the collected *Ginkgo biloba* samples, reflecting the high accuracy of MaxEnt for predicting the species geographic distribution. In addition, there are some unsuitable areas in the suitable area, which may be due to the influence of soil properties, elevation, or land use and human activities, making *Ginkgo biloba* unable to survive.

**Table 3.** The area of *Ginkgo biloba* suitable area under different scenarios ($10^4$ km$^2$).

| | Less Suitable Area | Moderately Suitable Area | Highly Suitable Area | Total Suitable Area |
|---|---|---|---|---|
| Current | 80.81 | 69.02 | 64.36 | 214.19 |
| 2050s_ssp126 | 72.21 | 79.85 | 121.99 | 274.05 |
| 2050s_ssp245 | 85.50 | 86.93 | 113.37 | 285.79 |
| 2050s_ssp370 | 68.16 | 68.40 | 144.43 | 280.99 |
| 2050s_ssp585 | 85.83 | 83.34 | 127.27 | 296.44 |
| 2070s_ssp126 | 86.75 | 75.04 | 85.92 | 247.71 |
| 2070s_ssp245 | 96.13 | 79.91 | 83.57 | 259.61 |
| 2070s_ssp370 | 84.90 | 97.02 | 144.96 | 326.87 |
| 2070s_ssp585 | 95.61 | 91.92 | 111.46 | 298.98 |

3.3.2. Simulation of Suitable Areas for *Ginkgo biloba* under Future Climate Change Scenarios and Its Change Analysis

Using ArcGIS10.6, we divided the probability maps of *Ginkgo biloba* existence under different scenarios simulated by MaxEnt to obtain geographic distribution patterns of *Ginkgo biloba* under different climatic conditions in the future periods (Figure 6). The results show the range of potential future suitable area as significant, increasing by the 2050s period and 2070s period under different climate scenarios, and the potential future distribution focused in most of the south and part of the northeast is the geographic space migrate from the current 24°–41° N, 102°–124° E to the future range of 23°–48° N, 93°–126° E. In addition, under the 2070s_ssp370 climate scenario (Table 3), the *Ginkgo biloba* total suitable area may reach a maximum area of $326.87 \times 10^4$ km$^2$, which increases by $112.68 \times 10^4$ km$^2$, and the growth ratio reaches 52.61%. While in the future 2070s_ssp126 climate scenario, the total suitable area maybe the smallest ($247.71 \times 10^4$ km$^2$), it also increased compared with the current total suitable area of *Ginkgo biloba* with an increase of $33.52 \times 10^4$ km$^2$ and a percentage of 15.56%. Then, based on the climate change, the highly suitable area tends to move northward (Figure 6), which might be because the increase in rainwater in the south inhibited the growth of *Ginkgo biloba*; on the contrary, increasing temperatures in the north promote its growth.

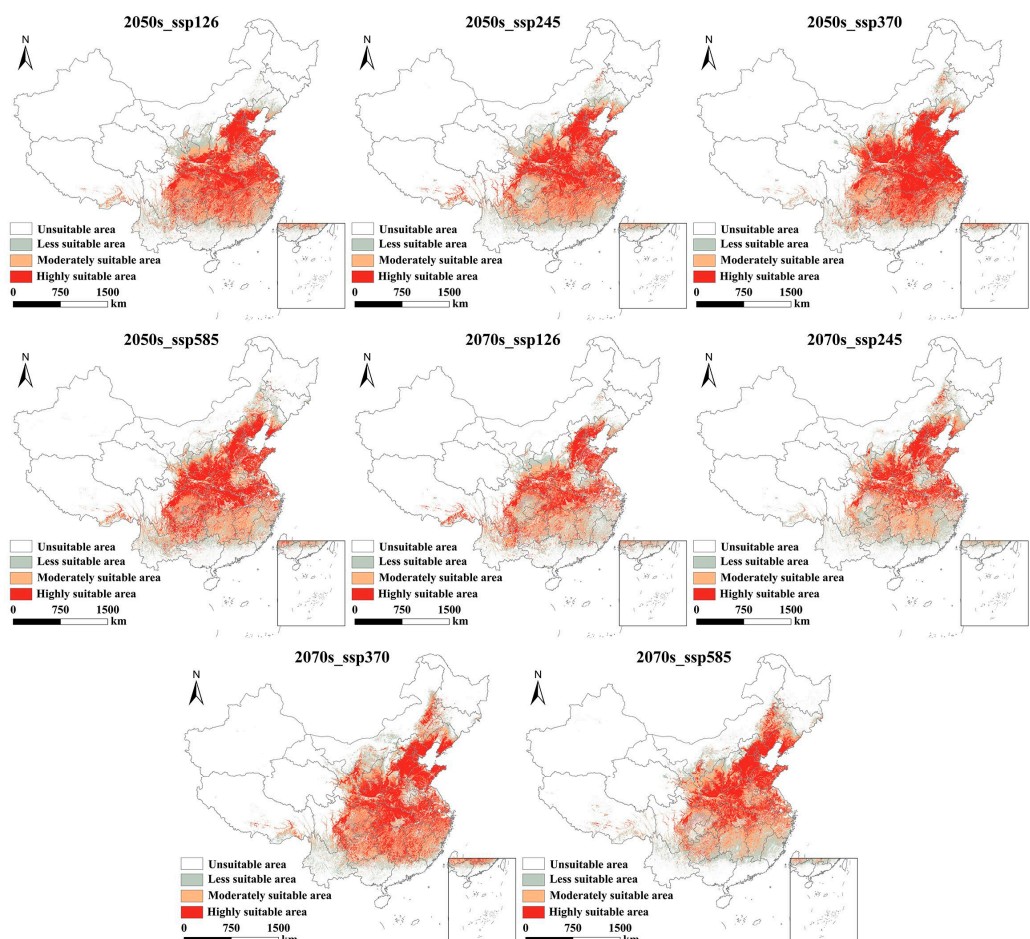

**Figure 6.** Distribution map of the *Ginkgo biloba* suitable area predicted under future climate scenarios.

Figure 7 shows the changes in the suitable area. Under different climate change scenarios, the *Ginkgo biloba* suitable area may expand compared to the current, and is relatively stable. By the future 2050s period, the total *Ginkgo biloba* suitable area may increase by 27.94%, 33.42%, 31.19% and 38.40% under the four climate scenarios, compared to the current one. By the 2070s period, under the four climate scenarios, the *Ginkgo biloba* suitable area may increase by 15.65%, 21.20%, 52.61% and 39.58%. From Figure 7, the *Ginkgo biloba* suitable area may gradually extend northward and westward in the future period under the influence of climate change, spreading to Yunnan Province in the west and expanding to the part of the area in Sichuan Province and Guangxi. The north may extend to the north, gradually crossing Shanxi Province and expanding to Ningxia, while in the northeast direction, it may extend to parts of Liaoning Province and expand as far as parts of the border of Neimenggu, Heilongjiang and Jilin. Although the changes in the *Ginkgo biloba* suitable area in the 2050s and 2070s period vary among scenarios, as shown in Figure 7, the extent of the *Ginkgo biloba* suitable area maintains a large degree of similarity in future distribution, and the common expansion areas of the *Ginkgo biloba* suitable area in each scenario are probably the north and northeastern parts of the current suitable area.

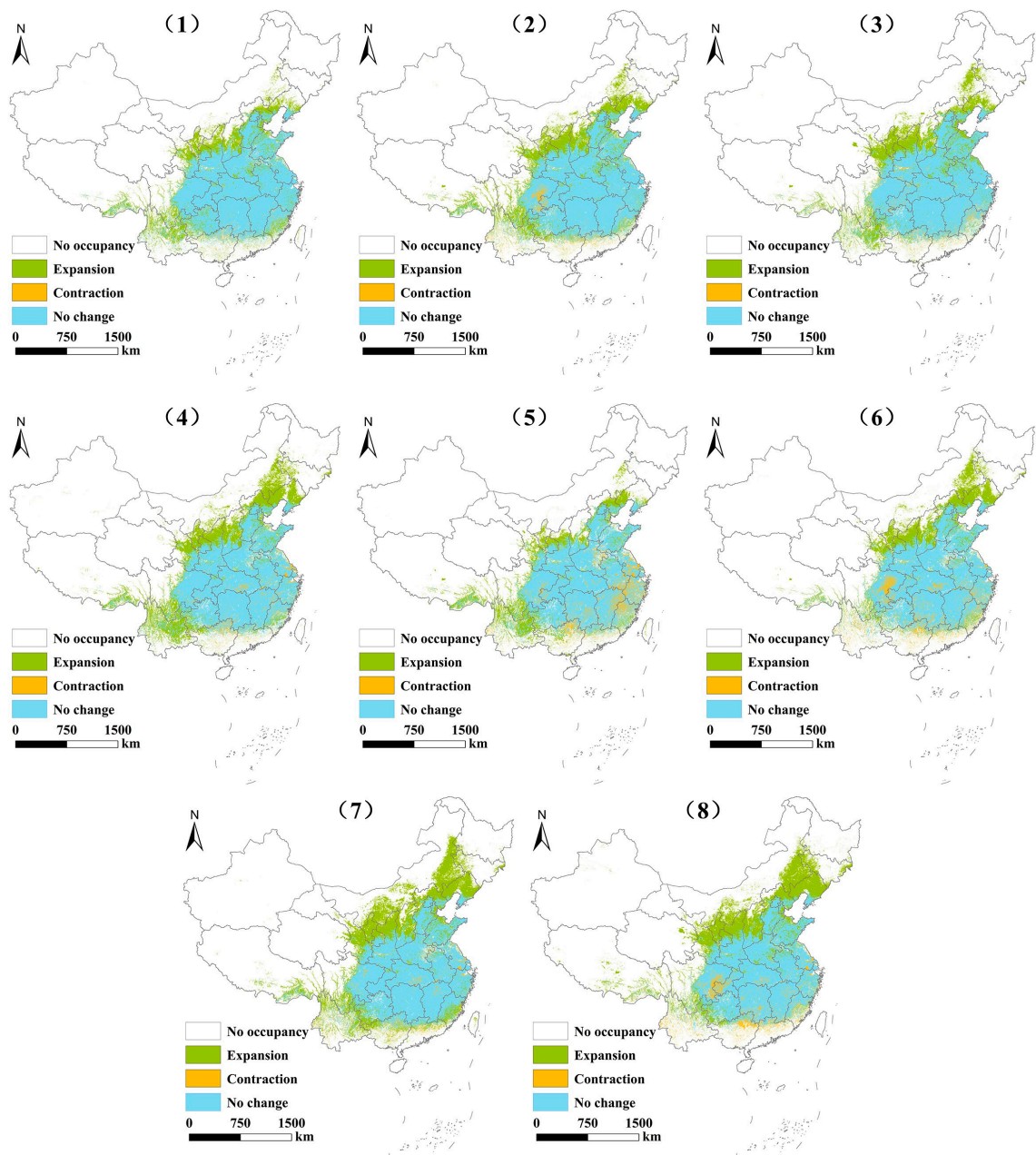

**Figure 7.** Possible changes of the suitable area of *Ginkgo biloba* in the future (Note: *Ginkgo biloba* suitable areas in different scenarios in the 2050s and 2070s period are compared with the current period: (**1**) Changes in the current period to the 2050s_ssp126 climate scenario; (**2**) Changes in the current period to the 2050s_ssp245 climate scenario; (**3**) Changes in the current period to the 2050s_ssp370 climate scenario; (**4**) Changes in the current period to the 2050s_ssp585 climate scenario; (**5**) Changes in the current period to the 2070s_ssp126 climate scenario; (**6**) Changes in the current period to the 2070s_ssp245 climate scenario; (**7**) Changes in the current period to the 2070s_ssp370 climate scenario; (**8**) Changes in the current period to the 2070s_ssp585 climate scenario).

According to the possible change in the *Ginkgo biloba* suitable area, the area of change in three types of suitable area expansion, contraction, and no change are counted, as shown in Table 4. From Table 4, it can be seen from that only a small part of the area change type of *Ginkgo biloba* may contract in the 2050s and 2070s. Most of the current suitable area may remain in the future, and on this basis, the area of the *Ginkgo biloba* suitable area may continue to expand. Compared with the current area of the *Ginkgo biloba* suitable area, by

the 2050s, under different climate scenarios, the *Ginkgo biloba* suitable area may expand by $63.91 \times 10^4$ km$^2$, $80.14 \times 10^4$ km$^2$, $75.87 \times 10^4$ km$^2$ and $91.04 \times 10^4$ km$^2$, and by the 2070s, the expansion of the *Ginkgo biloba* suitable area might be $55.44 \times 10^4$ km$^2$, $66.20 \times 10^4$ km$^2$, $120.17 \times 10^4$ km$^2$ and $102.76 \times 10^4$ km$^2$ under different climate scenarios, respectively. It reflects that the future global changes dominated by climate warming are favorable to the growth of *Ginkgo biloba*, although to a certain extent, it has caused limitations to the growth of *Ginkgo biloba*, but also makes *Ginkgo biloba* have a broader space for survival.

**Table 4.** The change area of the *Ginkgo biloba* suitable area under climate change and its proportion in the current *Ginkgo biloba* suitable area.

| | | Expansion ($\times 10^4$ km$^2$) | Proportion | Contraction ($\times 10^4$ km$^2$) | Proportion | No Change ($\times 10^4$ km$^2$) | Proportion |
|---|---|---|---|---|---|---|---|
| 2050s | ssp126 | 63.91 | 22.98% | 4.06 | 1.46% | 210.13 | 75.56% |
| | ssp245 | 80.14 | 27.23% | 8.55 | 2.90% | 205.65 | 69.87% |
| | ssp370 | 75.87 | 26.16% | 9.07 | 3.13% | 205.13 | 70.72% |
| | ssp585 | 91.04 | 29.83% | 8.79 | 2.88% | 205.40 | 67.29% |
| 2070s | ssp126 | 55.44 | 20.56% | 21.92 | 8.13% | 192.28 | 71.31% |
| | ssp245 | 66.20 | 23.61% | 20.79 | 7.41% | 193.40 | 68.98% |
| | ssp370 | 120.17 | 35.94% | 7.49 | 2.24% | 206.70 | 61.82% |
| | ssp585 | 102.76 | 32.42% | 17.97 | 5.67% | 196.22 | 61.91% |

### 3.4. Changes in the Center of Gravity of the Ginkgo biloba Suitable Area

To understand the trend of the *Ginkgo biloba* suitable area in the future, we used the mean center of gravity and standard deviation ellipse tool in ArcGIS 10.6 to obtain the possible center of gravity and direction of distribution of the *Ginkgo biloba* suitable area under each scenario (Figure 8). From Figure 8, the center of gravity of *Ginkgo biloba* distribution is currently in the northern part of Guizhou Province (107.5° E, 28.7° N). In the ssp126 scenario, the center of gravity of *Ginkgo biloba* distribution may move northeastward to the southwestern part of Hubei Province (108.7° E, 30.1° N) by the 2050s, and to the northern part of Hunan Province (109.9° E, 29.7° N) by the 2070s. Under the ssp245 scenario, the center of gravity of *Ginkgo biloba* distribution in the next two periods may move to the northeastern part of Chongqing City. Under the ssp370 scenario, the direction of movement of the *Ginkgo biloba* suitable area in the 2050s and 2070s might be approximately the same, with the difference that the center of gravity may move to central Chongqing City (107.8° E, 30.1° N) in the 2050s and to northeastern Sichuan Province (108.3° E, 31.9° N) in the 2070s. Under the ssp585 scenario, the center of gravity of the *Ginkgo biloba* suitable area in the 2050s may move approximately to the boundary of Hubei Province, Shanxi Province and Chongqing City (109.7° E, 31.8° N), and the center of gravity of the *Ginkgo biloba* suitable area in the 2070s may move to the southwestern part of Shanxi Province (107.4° E, 32.7° N). In summary, the center of gravity of *Ginkgo biloba* may shift by 2.4° in longitude and 4° in latitude under different climatic scenarios in the above two periods. The direction of their movement remains largely consistent, all moving northeastward. In addition, according to the standard deviation ellipse in Figure 8, the *Ginkgo biloba* suitable area may show a distribution from southwest to northeast, similar to the expansion direction of the *Ginkgo biloba* suitable area.

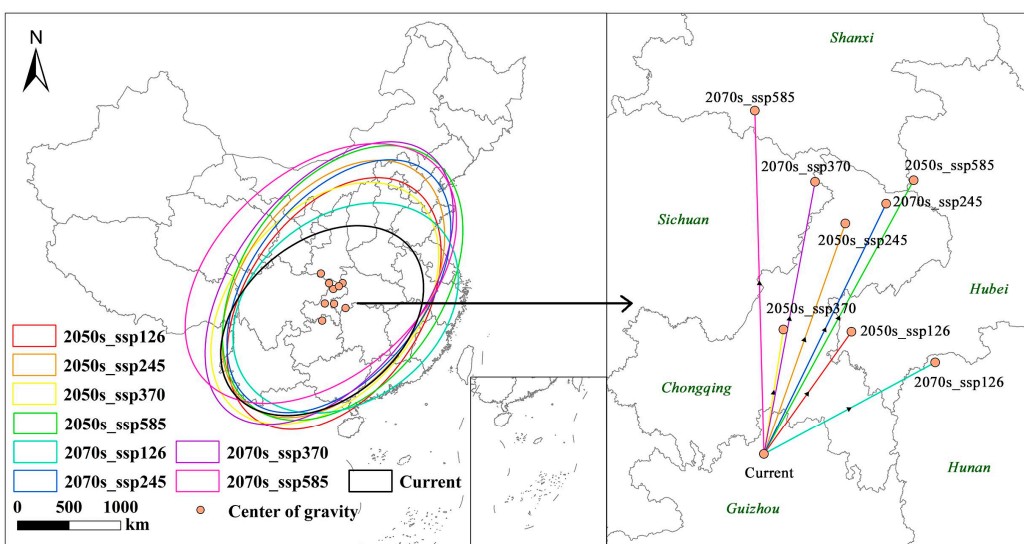

**Figure 8.** Changes in the distribution center of gravity of *Ginkgo biloba*.

## 4. Discussion

The MaxEnt model is applied to simulate and predict spatial changes in geographic distribution patterns of species and plays an active role in biogeography [55]. In this paper, we used the model MaxEnt 3.4.4 to simulate the distribution of the suitable area of *Ginkgo biloba*, and the simulation was very effective. According to the analysis of environmental variables evaluated by the Jackknife method, Bio6 was the main environmental variable, with a contribution of 43.5% in the model simulation, which is crucial to the distribution of *Ginkgo biloba*, followed by Bio13, with a contribution of 21.8%, which also plays a very important role in the distribution of the suitable area. By analyzing the dynamics of *Ginkgo biloba* suitable areas under different scenarios in the future period, it is favorable to the cultivation and planting of *Ginkgo biloba*. It is worth noting that *Ginkgo biloba* is stable in southeastern China, including eastern Sichuan, Chongqing, Hubei, Hunan, Jiangxi, Zhejiang, and northern Fujian, because the complex topography of these regions is prone to regional microclimates, which can mitigate climate change in the region and create stable climatic conditions for plants [59].

During periods of tree cultivation and growth, temperature and water utilization affect their physiological activities and biochemical processes [38]. *Ginkgo biloba* likes light and is suitable to grow in a natural forest with an altitude between 500 and 1000 m, a soil PH value between 5 and 5.5 and good drainage. It has high requirements for water and heat conditions; if the temperature is too high or too low, it will restrict the growth of *Ginkgo biloba*; as such, too much water will cause yellow leaves, dead branches or death of *Ginkgo biloba*, and too little water will seriously affect the quality and yield of *Ginkgo biloba* seeds. The threshold ranges of main environmental variables analyzed by the model in this paper reflect the important effects of temperature and precipitation on *Ginkgo biloba*, and these findings are consistent with the growth habits of *Ginkgo biloba*, reflecting the reliability of this paper. Under climate change, the predicted potential suitable area of *Ginkgo biloba* moved northward, indicating that the northern boundary of subtropical and temperate zones in China may therefore move northward. Some areas in the north have formed an environment suitable for the growth of *Ginkgo biloba* due to the changes in temperature. The topography, altitude, and geographical location of the Inner Mongolia Plateau and the Greater Khingan Mountains limits the growth of *Ginkgo biloba*, so the predicted potential suitable areas for *Ginkgo biloba* are distributed in the south. Next, the distribution of the *Ginkgo biloba* suitable area simulated in this paper has high similarity with that predicted by Guoying et al. [60–62] based on the IPCC Fifth Climate Assessment Report, and all conclude that temperature and precipitation have important influence on the *Ginkgo biloba* suitable area, which provides scientific support for the conclusions of this paper. Compared

with the studies of Guoying et al., in this paper, in addition to climatic variables, the effects of topography and soil properties on the growth of *Ginkgo biloba* are also considered, and a more scientific approach is adopted for the delineation of the suitable area distribution rather than simply based on the presence probability, which reflected that the research in this paper was more scientific.

In terms of ecological niche model selection, the paper combines current climate conditions with different greenhouse gas concentration pathways under climate conditions in two future periods (2050s and 2070s) through MaxEnt, in order to predict future distribution of *Ginkgo biloba*, which is widely used to predict species suitable area, but this does not mean that the predicted distribution of *Ginkgo biloba* is exactly the same as the actual distribution [27]. The MaxEnt model is uncertain in its simulations of species distributions, many studies have optimized the MaxEnt model by setting the feature classes and regularization multiplier used in the model training process [63,64]. The size of spatial resolution, the choice of environment variables, and the background range of variable data also affect the results of the model [51,65]; these aspects should be researched more in the future to optimize the model. The sampling deviation of the model background points caused by the obvious tendency of the sample distribution will also affect the prediction results [66], although it is very effective in simulating the distribution of species suitable areas using a small number of species sample data and limited information on environmental variables. However, these environmental variables cannot fully and accurately explain the current and future distribution of species. Moreover, the land use change and human disturbance factors are not taken into account in the modeling process, and these limitations can cause bias in the simulation results. Despite the many assumptions and uncertainties in species distribution simulations, the MaxEnt model is still an important data source tool for species distribution simulation predictions of future suitable areas, which is useful for proposing and developing scientific adaptation strategies based on predicted species distributions to cope with future climate change impacts on species, as well as ecosystem-level biota [67,68].

Global climate change has led to a variety of climatic and environmental issues that are already strongly affecting tree growth and vegetation community dynamics [69–71], which in turn affect tree growth and development through physiological stresses on temperature and precipitation [72]. As *Ginkgo biloba* is an endangered species, we studied the impact of climate change, which is of great practical significance for the conservation management and sustainable development of *Ginkgo biloba*. The potential geographical distribution pattern of *Ginkgo biloba*, predicted by the simulation, can be used as the base map for the protection planning of *Ginkgo biloba*. The main environmental variables affecting the growth of *Ginkgo biloba* obtained from the research results provide a reference for the artificial cultivation of *Ginkgo biloba*. In addition, *Ginkgo biloba* has a high economic, cultural, and ecological value in China and the world for its medicinal, ornamental, and timber uses. The study by Shareena et al. [71] showed that many *Ginkgo biloba* derivatives have been studied and elucidated to have significant therapeutic effects in many diseases. Therefore, studying the impact of future climates on the *Ginkgo biloba* suitable area has a high application value and can provide a scientific reference basis for *Ginkgo biloba*. conservation management and solving regional ecological and environmental problems, which further brings out the various values of *Ginkgo biloba* and plays a great role in economic development and human welfare. The future distribution map of *Ginkgo biloba* can help cultivation and planting of *Ginkgo biloba*, strengthen its management and protection, and serve as an important basis for future *Ginkgo biloba* development planning.

In addition, although the effects of climate, soil and topography on the *Ginkgo biloba* suitable area are considered in the ecological niche model in this paper, there are still limitations. In addition to being influenced by natural factors, the distribution of species has always been influenced by the important role of socioeconomic development, anthropogenic disturbances, policies and other human activities, and the role of various factors on species distribution should be considered comprehensively in future studies.

## 5. Conclusions

Under the influence of climate change, which is dominated by global warming, there are numerous restrictions on the range of species. Research and analysis on geographic distribution patterns of species and understanding its potential suitable areas for survival and the dynamics of species populations in different periods under climate change are important for the conservation of biodiversity under future climate conditions, as well as the conservation, management, and sustainable use of species [73]. In this paper, we simulated the current distribution pattern of *Ginkgo biloba* via the MaxEnt model and predicted the potential geographic distribution of *Ginkgo biloba* in the 2050s and 2070s through the data of *Ginkgo biloba* species distribution, bioclimatic, topographic and soil. The main environmental variables affecting the suitable area of *Ginkgo biloba* are also analyzed, and the distribution and changes of *Ginkgo biloba* suitable areas under different climatic scenarios are compared. We found that *Ginkgo biloba* suitable areas may expand in the future, spreading over most of the southern region and part of the northeastern region, and the highly suitable area of *Ginkgo biloba* may gradually shift northward. The variation in the suitable area for different classes shows that moderate temperature and precipitation changes under climate change will promote the growth of *Ginkgo biloba* In the future period, *Ginkgo biloba* suitable areas may show a distribution pattern from southwest to northeast, and the center of gravity of distribution may move to the northeast.

The ecological, cultural, and economic values of *Ginkgo biloba* have rich connotations, and *Ginkgo biloba*-related products have very broad application prospects. However, so far, only wild *Ginkgo biloba* remain in China, while all other countries cultivate *Ginkgo biloba* by artificial cultivation. However, in China, besides the existing wild *Ginkgo biloba*, many cultivated *Ginkgo biloba* are needed to meet the demand of society, and the conservation and management of *Ginkgo biloba* should be closely monitored. The research results above intend to provide theoretical guidance for artificial cultivation of *Ginkgo biloba* under climate change. Next, they provide a scientific basis for the conservation management of wild *Ginkgo biloba* and its related ecological and environmental issues, as well as facilitating the sustainable use of *Ginkgo biloba* resources and the conservation of endangered species.

In this regard, based on the predicted *Ginkgo biloba* distribution map and the main environmental variables affecting the growth of *Ginkgo biloba*, effective measures should be taken in areas with favorable climatic conditions to achieve targeted protection of the *Ginkgo biloba*. First, it is necessary to establish natural-wild *Ginkgo biloba* protected areas, which is the main way to protect endangered species. The protected areas should give priority to the areas where the current climate scenario overlaps with the potential suitable areas under the future climate scenario. And then, when selecting the site of artificial cultivation of *Ginkgo biloba* to meet people 's needs, it can also be selected according to the potential distribution range of *Ginkgo biloba*, and the effects of temperature, precipitation, and other factors on the growth of *Ginkgo biloba* should also be considered.

**Supplementary Materials:** The following supporting information can be downloaded at: https://www.mdpi.com/article/10.3390/f14122284/s1, File S1. Note of 19 bioclimatic variables.

**Author Contributions:** Conceptualization, J.Z.; methodology and validation, Y.H.; software, J.Z. and Y.H.; formal analysis, L.T.; investigation, Y.Z. and L.T.; writing—original draft preparation, review and editing, Y.Z. and C.S.; visualization, L.T.; supervision, project administration and funding acquisition. All authors have read and agreed to the published version of the manuscript.

**Funding:** This research was funded by the Basic Frontier Science Research Program of the Chinese Academy of Sciences Original innovation projects from 0 to 1 (ZDBS-LY-DQC023), the second Tibetan Plateau Scientific Expedition Program (2019QZKK0608), and the Belt and Road Special Foundation of the State Key Laboratory of Hydrology-Water Resources and Hydraulic Engineering (2021490111).

**Data Availability Statement:** The data is available on request from the corresponding author.

**Conflicts of Interest:** The authors declare no conflict of interest.

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
