# Peer review of "The Ginkgo biloba L. in China: Current Distribution and Possible Future Habitat"

_forests, doi:10.3390/f14122284_

Round 1

Reviewer 1 Report

Comments and Suggestions for Authors

few minor grammatical and typo errors (highlighted in yellow with comment)

Author Response

Reviewer #1

Few minor grammatical and typo errors (highlighted in yellow with comment).

Response: Thank you for your comments. We have modified the syntax problem you pointed out (corresponding to lines 24-25, 30-32, 32-35 of the revised file), specific as follows. We corrected the correct expression of “Ginkgo biloba L.” and modified the font form (red font of the revised file).

Lines: 24-25

It not only can produce economic and ecological benefits, but also produce huge social benefits.

Lines: 30-32

(2) The Min Temperature of Coldest Month, the Precipitation of Wettest Month, elevation, and Temperature Seasonality are the main environmental variables affecting the change of Ginkgo biloba L. suitable area.

Lines: 32-35

“(3) Under the future climate scenarios, the suitable area of Ginkgo biloba L. is predicted to expand in future, covering most of the south and some northeast regions, and moderate temperature and precipitation changes under climate change are conducive for the growth of Ginkgo biloba L.

Reviewer 2 Report

Comments and Suggestions for Authors

I believe that it is an effective research on the importance of Ginkgo biloba and its estimated distribution in the future. 

42. Line; The stated temperature increase amounts are assumptions and the maximum increase probability expression would be more accurate.

74. line: I could not find a statement in the reference given that the Max Ent program is a statistical program that gives the optimal result. Instead, the reason for preference should be expressed better.

Discussion

The climatic and topographic differences of the Northern region can be discussed here.

Thank you so much.

Author Response

I believe that it is an effective research on the importance of Ginkgo biloba and its estimated distribution in the future.

Comment 1: 42. Line; The stated temperature increase amounts are assumptions and the maximum increase probability expression would be more accurate.

Response: Thank you for your comments. According to your advice, we have made changes to the statement expression (corresponding to lines 43-45 of the revised file). Specific as follows.

Lines: 43-45

The IPCCs sixth assessment report (AR6) noted that global temperatures are likely to rise by a maximum of 0.7-4.0 °C relative to 1995-2014 around 2100 years

Comment 2: 74. line: I could not find a statement in the reference given that the Max Ent program is a statistical program that gives the optimal result. Instead, the reason for preference should be expressed better.

Response: Thank you for your comments. The advantages of MaxEnt in simulating species distribution have been described in the text (corresponding to lines 70-77and 549-550 of the revised file), details can be found in reference [21]. Specific as follows.

Line: 70-77

At present, MaxEnt model is the optimal solution of species distribution model in terms of accuracy and stability [22]. Its accuracy is higher than other species distribution models, and it is widely used to predict species potential distribution pattern [23]. It can fit species distribution area by known sample points and corresponding environmental variables with maximum entropy and predict the probability of species existence under the influence of environmental variables [21]. Moreover, MaxEnt is suitable for some cases where the data information is not clear [24] and is therefore suitable for simulating species distribution [25-26].

Lines: 549-550

21. Phillips, S.J.; Anderson, R.P.; Schapire, R.E. Maximum entropy modeling of species geographic distributions. Ecol. Modell. 2006, 190, 231-259. [CrossRef]

Comment 3: Discussion, The climatic and topographic differences of the Northern region can be discussed here.

Response: Sincerely thank you for your valuable comments. We have increased our discussion of climate and topography (corresponding to lines 371-375 and 385-392 of the revision file). The details are as follows.

Lines: 371-375

“It is worth noting that Ginkgo biloba L. is stable in southeastern China, including eastern Sichuan, Chongqing, Hubei, Hunan, Jiangxi, Zhejiang, and northern Fujian, because the complex topography of these regions is prone to regional microclimates, which can mitigate climate change in the region and create stable climatic conditions for plants [59].”

Lines: 385-392

“Under climate change, the predicted potential suitable area of Ginkgo biloba L. moved northward obviously, indicating that the northern boundary of subtropical and temperate zones in China may move northward. Some areas in the north have formed an environment suitable for the growth of Ginkgo biloba L. due to the changes in temperature. The topography, altitude, and geographical location of the Inner Mongolia Plateau and the Greater Khingan Mountains limits the growth of Ginkgo biloba L., so the predicted potential suitable areas for Ginkgo biloba L. are distributed in their south.”

Reviewer 3 Report

Comments and Suggestions for Authors

Dear Authors

 I recently read your paper titled “The Ginkgo Trees in China—Current Distribution and Possible Future Habitat” with great interest, and I appreciate the valuable insights your study contributes to understanding Ginkgo tree distribution in the context of climate change. Your work has prompted several questions and considerations, and I am reaching out to seek clarification and additional insights on various aspects of your research. Here are some specific points I would appreciate your input on:

 1. Environmental Variable Selection:

   - Could you provide more details on the rationale behind selecting the specific environmental variables used in your study? How were these variables chosen, and were there considerations for excluding other potentially relevant factors?

 2. Model Validation:

   - While AUC values are presented for model accuracy, could you elaborate on the validation methods employed? Did you consider cross-validation or testing the model on an independent dataset to ensure its robustness?

 3. Impact of Non-climatic Factors:

   - The paper briefly mentions non-climatic factors such as soil properties, elevation, and land use. How were these factors incorporated into the MaxEnt model, and to what extent do they contribute to the prediction of suitable areas for Ginkgo trees?

 4. Ginkgo Tree Species Variability:

   - Ginkgo trees can exhibit variability in their responses to environmental changes. Did your analysis consider different Ginkgo species or populations, and if so, how was this variability addressed in the modeling process?

 5. Future Climate Scenarios:

   - What was the rationale behind choosing the specific greenhouse gas emission scenarios (ssp126, ssp245, ssp370, ssp585)? Did you consider uncertainties in these scenarios, and why did you focus on the 2050s and 2070s time frames?

 6. Discussion of Model Limitations:

   - While the advantages of the MaxEnt model are discussed, could you provide a more detailed discussion on its limitations? For instance, how sensitive is the model to changes in input data or parameter settings?

 7. Conservation Implications:

   - The paper mentions providing a theoretical basis for the protection and management of Ginkgo trees. Could you elaborate on the specific conservation implications or recommendations that can be drawn from your study?

 8. Comparison with Previous Studies:

   - If applicable, how do your findings compare with or differ from previous studies on Ginkgo tree distribution or similar species distribution modeling?

 I appreciate your time and expertise, and I believe your responses will enhance the understanding of your research within the broader scientific community. Thank you for your consideration, and I look forward to hearing from you.

Comments on the Quality of English Language

After reviewing the manuscript, I suggest that a minor editorial review is needed to enhance the overall quality of the English language.

Author Response

I recently read your paper titled “The Ginkgo Trees in China—Current Distribution and Possible Future Habitat” with great interest, and I appreciate the valuable insights your study contributes to understanding Ginkgo tree distribution in the context of climate change. Your work has prompted several questions and considerations, and I am reaching out to seek clarification and additional insights on various aspects of your research. Here are some specific points I would appreciate your input on:

Comment 1: Environmental Variable Selection: - Could you provide more details on the rationale behind selecting the specific environmental variables used in your study? How were these variables chosen, and were there considerations for excluding other potentially relevant factors?

Response: Sincerely thank you for your comments. In the process of selecting environmental variables, we comprehensively considered the factors affecting plant growth, such as climate, topography, soil properties, and land use. However, because some data are difficult to obtain in the future, we finally chose bioclimate, soil, and topography with reference to relevant literature (corresponding to lines 128-132 and 585-586 of the revised file). The limitations of some other relevant factors, such as human activities, land use change, etc., are explained in the discussion (corresponding to lines 446-451 of the revised file). Specific as follows.

Lines: 128-132

Plant growth is closely related to climate, topography, soil properties, vegetation types, NDVI and so on. In this study, the current and future distribution patterns of Ginkgo biloba L. will be simulated. Considering the availability of data in these periods (such as NDVI data in the future period cannot be obtained), we selected three environmental variables: bioclimate, soil and topography [38].

Lines: 585-586

38. Zhao, H.R, Yang, X.T., Shi, S.Y., Xu, Y.D., Yu, X.P., Ye, X.P. Climate-driven distribution changes for Bashania fargesii in the Qinling Mountains and its implication for panda conservation. Glob. Ecol. Conserv. 2023. 46, e02610. [CrossRef]

Lines: 446-451

In addition, although the effects of climate, soil and topography on Ginkgo biloba L. suitable area are considered in the ecological niche model in the paper, there are still limitations. In addition to being influenced by natural factors, the distribution of species has always been influenced by the important role of socioeconomic development, anthropogenic disturbances, policies and other human activities, and the role of various factors on species distribution should be considered comprehensively in future studies.

Comment 2: Model Validation: - While AUC values are presented for model accuracy, could you elaborate on the validation methods employed? Did you consider cross-validation or testing the model on an independent dataset to ensure its robustness?

Response: Sincerely thank you for your comments. In the process of modeling, by evaluating the setting of the regularization multiplier value and the feature class, we set the combination with the smallest AICc as the best model (corresponding to lines 176-178 of the revised file). Then we set the Replicated run type as Bootstrap replicates, set 30% of the sample data as the test set to verify the accuracy of the training set according to reference [47] (corresponding to lines 178-180 and 604-605 of the revised file). According to the number of occurrences, we use cross-validation method to evaluate the model, set up repeated runs for 15 times, and use different sample points as test sets in the process of multiple repetitions to obtain more accurate model performance evaluation results (corresponding to lines 184-186 of the revised file). The AUC values of the model runs show that the model passed the accuracy validation and has reliability (corresponding to line 213 of the revised file). Specific as follows.

Lines: 176-178

Through the setting evaluation of regularization multiplier value and feature class. We set the minimum combination of AICc (Akaike Information Criterion) as the best model [46].”

Lines: 178-180

The Bootstrap replicates are set to 15, and 70% of the sample data are randomly selected as training set for model establishment. The remaining 30% are used as the test set [47],

Lines: 604-605

“47. Fang, B.; Zhao, Q.; Qin, Q.L.; Yu, J. Prediction of Potentially Suitable Distribution Areas for Prunus tomentosa in China Based on an Optimized MaxEnt Model. Forests. 2022, 13, 381. [CrossRef]”

Lines: 184-186

so according to the number of occurrences, the cross-validation method was used for model evaluation, and the number of repeated training is set to 15, so that the final output results tended to be balanced and reliable.

Line: 213

Table 2. The AUC values of simulation results

AUC of Training date

AUC of Test date

AUC of Random prediction

Current

0.9402

0.9050

0.5

2050s_ssp126

0.9421

0.9040

0.5

2050s_ssp245

0.9411

0.9124

0.5

2050s_ssp370

0.9390

0.9090

0.5

2050s_ssp585

0.9421

0.9080

0.5

2070s_ssp126

0.9425

0.9040

0.5

2070s_ssp245

0.9446

0.9120

0.5

2070s_ssp370

0.9404

0.9060

0.5

2070s_ssp585

0.9383

0.9001

0.5

Comment 3: Impact of Non-climatic Factors: - The paper briefly mentions non-climatic factors such as soil properties, elevation, and land use. How were these factors incorporated into the MaxEnt model, and to what extent do they contribute to the prediction of suitable areas for Ginkgo trees?

Response: Thank you for your comments. The non-climatic factors of soil and topography are the same as the climatic factors. The raster data obtained was processed and converted into an ASC format that can be read by MaxEnt, and then imported into MaxEnt for analysis together with the climate factors.

Comment 4: Ginkgo Tree Species Variability: - Ginkgo trees can exhibit variability in their responses to environmental changes. Did your analysis consider different Ginkgo species or populations, and if so, how was this variability addressed in the modeling process?

Response: Sincerely thank you for your comments. Ginkgo biloba L. is a relict plant, the only species of the genus Ginkgo in the family Ginkgoaceae, and natural wild Ginkgo biloba L. only exists in China, so there is no need to consider different ginkgo species or populations in the analysis process. These have been explained in the text (corresponding to lines 97-104 and 582-584 of the revised file). Specific as follows.

Lines: 97-104

In addition, although Ginkgo is common, it is actually an ancient relict plant in the world [35]. It is the only surviving organism of the genus Ginkgo in the family Ginkgoaceae and is susceptible to dramatic climate changes. Many years ago, Ginkgo biloba L. was extinct in most parts of Europe, North America, and Asia due to climate disturbance. Only Chinas preservation continues to this day, while most of the existing Ginkgo biloba L. in other countries is directly or indirectly introduced from China [36]. Due to the uniqueness of Ginkgo biloba L., it was included in the IUCN Red List of Endangered Species in 1998 [37].

Lines: 582-584

35. Fu, L.M.; Chin, C.M. China plant red data book: rare and endangered plants. Science Press. 1992. [CrossRef]

  1. Zhao, Y.P.; Paule, J.; Fu, C.X.; Koch, M.A. Out of China: Distribution history of Ginkgo biloba L. Taxon. 2010, 59, 495-504. [CrossRef]
  2. Sun, W. Ginkgo biloba. The IUCN Red List of Threatened Species 1998. China. 1998. [CrossRef]

Comment 5: Future Climate Scenarios: - What was the rationale behind choosing the specific greenhouse gas emission scenarios (ssp126, ssp245, ssp370, ssp585)? Did you consider uncertainties in these scenarios, and why did you focus on the 2050s and 2070s time frames?

Response: Thank you for your comments. We chose to study the potential spatial distribution of Ginkgo biloba L. under climate change because it is an endangered species and it is specialized, with high medicinal, practical, ornamental, and research values. Based on the relevant literature, we chose the four scenarios SSP126, SSP245, SSP370 and SSP585 from low to high in order to enrich the results of the study by reflecting the possible future distribution of Ginkgo biloba L. suitable areas under the influence of different concentrations of greenhouse gases, which can give more inspiration to you. The scenario setting has been widely used by many scholars. The BCC-CSM2-MR model we selected has greatly improved the simulation ability of China 's climate and is suitable for the research area of this paper. In addition, we want to provide a reference for the protection and management of Ginkgo biloba L.'s future sustainable development through this article. The time span cannot be too close or too far from now, so as to have the time to make a plan, so we chose the 2050s. It is also important to reflect trends that are likely to develop in the long term under a given climate scenario, so we also chose the 2070s (corresponding to lines 132-143 and 587-594 of the revised file). Specific as follows.

Lines: 132-143

The bioclimatic variables data (1970-2000, 2041-2060/2050s period, 2061-2080/2070s period) in this paper are derived from the WorldClim 2.1 (http://www.worldclim.org/) CMIP6 model data. Compared with the CMIP5 version of the climate model, the simulation ability of the BCC-CSM2-MR for Chinas climate has been greatly improved [39]. Therefore, the future bioclimatic variable factors were selected under the BCC-CSM2-MR module, four different concentrations of greenhouse gas emission scenarios (ssp126 is a low forcing scenario, ssp245 is a medium forcing scenario, ssp370 is a medium to high forcing scenario, and ssp585 is a high forcing scenario [40-41]) in the 2050s period (2041-2060 average) and the 2070s period (2061-2080 average). These two future periods are relatively suitable for the time span from the current period. It is convenient to take corresponding biological protection measures in time according to the prediction results.”

Lines: 587-594

39. Wu, T., Lu, Y., Fang, Y., Xin, X., Li, L., Li, W., Jie, W., Zhang, J., Liu, Y., Zhang, L., Zhang, F., Zhang, Y., Wu, F., Li, J., Chu, M., Wang, Z., Shi, X., Liu, X., Wei, M., Huang, A., Zhang, Y., and Liu, X.: The Beijing Climate Center Climate System Model (BCC-CSM): the main progress from CMIP5 to CMIP6. Geosci. Model Dev. 2019. 12, 15731600. [CrossRef]

  1. O'Neill, B.; Tebaldi, C.; Vuuren, D.; Eyring, V.; Friedlingstein, P.; Hurtt, G.; Knutti, R.; Kriegler, E.; Lamarque, J-F.; Lowe, J.; Meehl, G.; Moss, R.; Riahi, K.; Sanderson, B. The Scenario Model Intercomparison Project (ScenarioMIP) for CMIP6. Geosci. Model. Dev. 2016, 9, 3461-3482. [CrossRef]
  2. Huang, Y.M., Zhang, G.L., Fu, W.D., Zhang, Y., Zhao, Z.H., Li, Z.H., Qin, Y.J. (2023). Impacts of climate change on climatically suitable regions of two invasive Erigeron weeds in China. Front. Plant Sci. 2023. 14. [CrossRef]

Comment 6: Discussion of Model Limitations: - While the advantages of the MaxEnt model are discussed, could you provide a more detailed discussion on its limitations? For instance, how sensitive is the model to changes in input data or parameter settings?

Response: Sincerely thank you for your comments. We discuss the limitations and parameter settings of the MaxEnt model in the fourth part (corresponding to lines 407-414, 612-613 and 637-645 of the revised file). Specific as follows.

Lines: 407-414

The MaxEnt model is uncertain in its simulations of species distributions, many studies have optimized the Maxent model by setting the feature classes and regularization multiplier used in the model training process [63-64]. The size of spatial resolution, the choice of environment variables, and the background range of variable data also affect the results of the model [51,65], these aspects should be researched more in the future to optimize the model. The sampling deviation of the model background points caused by the obvious tendency of the sample distribution will also affect the prediction results [66].

Lines: 612-613

51. Şen, İ.; Sarıkaya, O.; Örücü, Ö.K. Predicting the future distributions of Calomicrus apicalis Demaison, 1891 (Coleoptera: Chrysomelidae) under climate change. J. Plant Dis. Protect. 2022, 129, 325-337. [CrossRef]

Lines: 637-645

63. Khan, A.M., Li, Q.T, Saqib, Z., Khan, N., Habib, T., Khalid, N., Majeed, M., Tariq, A. MaxEnt Modelling and Impact of Climate Change on Habitat Suitability Variations of Economically Important Chilgoza Pine (Pinus gerardiana Wall.) in South Asia. Forests. 2022, 13(5). [CrossRef]

  1. Amaro, G., Fidelis, E.G., da, S.R.S., Marchioro, C.A. Effect of study area extent on the potential distribution of Species: A case study with models for Raoiella indica Hirst (Acari: Tenuipalpidae). Ecol. Modell. 2023, 483, 110454. [CrossRef]
  2. Gong, H.D., Cheng, Q.P., Jin, H.Y., Ren, Y.T. Effects of temporal, spatial, and elevational variation in bioclimatic indices on the NDVI of different vegetation types in Southwest China. Ecol. Indic. 2023. 154, 110499. [CrossRef]
  3. Alatawi, A.S., Gilbert, F., Reader, T. Modelling terrestrial reptile species richness, distributions and habitat suitability in Saudi Arabia. Journal of Arid Environments. 2020. 178, 104153. [CrossRef]”

Comment 7: Conservation Implications: - The paper mentions providing a theoretical basis for the protection and management of Ginkgo trees. Could you elaborate on the specific conservation implications or recommendations that can be drawn from your study?

Response: Thank you for your comments. According to your suggestions, we have added measures for the protection and management of Ginkgo biloba L. in the article (corresponding to lines 482-491 of the revised file). Specific as follows.

Lines: 482-491

In this regard, based on the predicted Ginkgo biloba L. distribution map and the main environmental variables affecting the growth of Ginkgo biloba L., effective measures should be taken in areas with favorable climatic conditions to achieve targeted protection of the ginkgo. First, it is necessary to establish a natural wild Ginkgo biloba L. protected areas, which is the main way to protect endangered species. The protected areas should give priority to the areas where the current climate scenario overlaps with the potential suitable areas under the future climate scenario. And then, when selecting the site of artificial cultivation of Ginkgo biloba L. to meet people 's needs, it can also be selected according to the potential distribution range of Ginkgo biloba L., and the effects of temperature, precipitation, and other factors on the growth of Ginkgo biloba L. should also be considered.

Comment 8: Comparison with Previous Studies: - If applicable, how do your findings compare with or differ from previous studies on Ginkgo tree distribution or similar species distribution modeling?

Response: Thank you for your comments. Based on your suggestion, we added a comparison of our results with the distribution of Ginkgo biloba L. under climate change in previous studies (corresponding to lines 392-401 and 631-636 of the revised file). Specific as follows.

Lines: 392-401

Next, distribution of Ginkgo biloba L. suitable area simulated in this paper has high similarity with that predicted by Guoying et al [60-62] based on the IPCC Fifth Climate Assessment Report, and all conclude that temperature and precipitation have important influence on Ginkgo biloba L. suitable area, which provides scientific support for the conclusions of this paper. Compared with the studiies of Guoying et al., in this paper, in addition to climatic variables, the effects of topography and soil properties on the growth of Ginkgo biloba L. are also considered, and a more scientific approach is adopted for the delineation of the suitable area distribution rather than simply based on the presence probability, which reflected that the research in this paper was more scientific.

Lines: 631-636

60. Guo, Y. Prediction of ginkgo distribution and research on environmental response mechanism of phenotypic traits under the climate change. J.Nanjing Forestry Uni. 2021. [CrossRef]

  1. Liu, J. Historical Distribution and Migration Trend of Relict Plant Ginkgo biloba Based on Random Forest Model. Shaanxi Normal University: Xian, China. 2019. [CrossRef]

62.   Xu, Y. Prediction of Suitable Areas of Relict Plants Ginkgo biloba L. and Davidia Involucrata Based on Maximum Entropy Model. North China Electric Power University: Beijing, China. 2019. [CrossRef]”

Reviewer 4 Report

Comments and Suggestions for Authors

Comments on the Quality of English Language

Dear Editor

Please see my comments in the attached file above.

Regards

Author Response

The authors targeted to explore and quantify the potential habitat suitability variations of Ginkgo tree under different climatic conditions from China. They used the maximum entropy tool to perform distribution modelling of this tree species. Overall, this MS is fine and have plenty of useful information. However, the reproducibility of work presented in this MS isn’t possible because of some very important missing information. These information are particularly linked to M&M and some presentation issues that need to be addressed. My detail comments and recommendations are listed below.

Comment 1: Line 21. This MS need a thorough language editing. e.g. “the vast majority of wild Ginkgo trees is gradually extinct,” such phrases are needed to be revised for better clarity like “The China’s wild Ginkgo population is facing extinction risk…..”.

Response: Thank you for your comments. We re-explain this in the text (corresponding to lines 22-23 of the revised file). The details are as follows.

Lines: 22-23

“only China's wild ginkgo has been preserved, yet the Chinas wild Ginkgo biloba L. population is facing extinction risk”

Comment 2: Line 30-31. Rephrasing required in “Under the future climate scenarios, Ginkgo trees suitable area will expand”. The authors should treat this work as a predictive work and write/communicate accordingly. e.g. “Under the future climate scenarios, the habitat suitability of Ginkgo might expand” or “The habitat suitability of Ginkgo tree is predicted to expand in future”. Please revise your entire MS accordingly.

Response: Thank you for your comments, we modify the expression in the text to a more appropriate expression according to your suggestion. Due to the large number of modifications (mainly concentrated in section 3.3.2 and 3.4.), it is not possible to list them one by one. We list some here, you can see all in the revised file.

Lines: 32-35

(3) Under the future climate scenarios, the suitable area of Ginkgo biloba L. is predicted to expand in future, covering most of the south and some northeast regions, and moderate temperature and precipitation changes under climate change are conducive for the growth of Ginkgo biloba L.

Lines: 282-287

In addition, under the 2070s_ssp370 climate scenario (Table 3), Ginkgo biloba L. total suitable area may reach the maximum area of 326.87×104 km2, which increases by 112.68×104 km2, and the growth ratio reaches 52.61%. While in the future 2070s_ssp126 climate scenario, the total suitable area maybe the smallest (247.71×104 km2), it also increased compared with the current total suitable area of Ginkgo biloba L. with an in-crease of 33.52×104 km2 and a percentage of 15.56%.

Lines: 345-349

Under the ssp370 scenario, the direction of movement of the Ginkgo biloba L. suitable area in the 2050s and 2070s periods maybe approximately the same, with the difference that the center of gravity may move to central Chongqing City (107.8°E, 30.1°N) in the 2050s period and to northeastern Sichuan Province (108.3°E, 31.9°N) in the 2070s period.

Comment 3: You removed the duplicate occurrence points and wrong geographical distribution, but no information related to environmental filtering of presence points (or points thinning) are conveyed…Has you selected one presence point per grid/pixel to carry out SDM?

Response: Sincerely thank you for your comments. The data of Ginkgo biloba L. distribution points we used come from two species sharing platforms. The point data downloaded from these two platforms have repeated coordinate information partly, and we only retain one of them. Then, a few points are located in the sea of eastern China, we also remove them. In addition, since the resolution of the variable data involved in the analysis is 1 km, we only retain one of the two sample points with a ground distance of less than 2 km, we've added a description of this part in the text (corresponding to lines 121-123 of the revised file). Finally, based on these filtered points, we used the MaxEnt model to establish the relationship between Ginkgo biloba L. samples and surrounding environmental variables for simulation prediction analysis. In the process of processing the sample points, the filtered each sample point exist in a separate grid/pixel. The details are as follows.

Lines: 121-123

Then, due to the 1 km resolution of the variable data involved in the analysis, we retained only one of the two sample points with a ground distance of less than 2 km by buffer analysis.

Comment 4: Optimal MaxEnt model settings are not assessed and communicated. Use of varying FC and RM values significantly influence model output and should be used to identify a threshold-dependent (i.e., omission rate) evaluation metrics, in order to obtain optimal model settings, prevent overfitting and improve model transferability. (See https://doi.org/10.3390/f13050715).

Response: Sincerely thank you for your comments. In the process of modeling, by evaluating the setting of the regularization multiplier value and the feature class, we set the combination with the smallest AICc as the best model (corresponding to lines 176-178 and 602-603 of the revised file). Then we set the Replicated run type as Bootstrap replicates, set 30% of the sample data as the test set to verify the accuracy of the training set according to reference [47] (corresponding to lines 604-605 of the revised file), and repeat the run for 15 times, and in the process of repeated cross validation, different sample points are used as the test set, which can get more accurate model simulation results. Specific as follows.

Lines: 176-180

Through the setting evaluation of regularization multiplier value and feature class. We set the minimum combination of AICc (Akaike Information Criterion) as the best model [46]. The Bootstrap replicates are set to 15, and 70% of the sample data are randomly selected as training set for model establishment. The remaining 30% are used as the test set [47],”

Lines: 602-603

“46. Radosavljević, A., Anderson, R.P. Making better Maxent models of species distributions: complexity, overfitting and evaluation. J. Biogeogr. 2014. 41. [CrossRef]”

Lines: 604-605

“47. Fang, B.; Zhao, Q.; Qin, Q.L.; Yu, J. Prediction of Potentially Suitable Distribution Areas for Prunus tomentosa in China Based on an Optimized MaxEnt Model. Forests. 2022, 13, 381. [CrossRef]”

Comment 5: The extent of background environment while performing current distribution modelling is very important in SDM. The occurrence points (Fig. 2) mapping shows that the tree species is approximately distributed in about 1/3 land area of China, however, the authors probably used the entire china extent to clip the environmental layers to be used as model background environment. Different background extent significantly affect the model output and give an unrealistically high AUC value. (Please see; https://doi.org/10.1016/j.ecolmodel.2023.110454). Secondly, what about the selected number of background points? A poor background sampling may lead to a truncated environmental response.

Response: Sincerely thank you for your comments. We 've read the literature you 've given and learned about the significant impact of the study area on predicting the potential distribution of species. This paper mainly studies the potential distribution of Ginkgo biloba L. in China and their future potential suitable areas in China under climate change. At present, wild Ginkgo biloba L. only exists in China [36] (corresponding to lines 583 of the revised file), and the impact of other regions is small, so we use the whole China as the background environment of the model. For the number of background points, refer to the related literature [10], [29], [47], etc.to set. (corresponding to lines 180-181, 528-529, 567-568 and 604-605 of the revised file), many previous studies have proved that this setting is effective. Specific as follows.

Line: 583

36. Zhao, Y.P.; Paule, J.; Fu, C.X.; Koch, M.A. Out of China: Distribution history of Ginkgo biloba L. Taxon. 2010, 59, 495-504. [CrossRef]

Lines: 180-181

The remaining 30% are used as the test set [47], the maximum number of background points was 10,000 [10,29,47].”

Lines: 528-529

10. Lu, Z.Y.; Zhai, Y.P.; Meng, D.R.; Kou, G.Q.; Li, H.; Liu, J.Z. Predicting the potential distribution of wintering Asian Great Bustard (Otis tarda dybowskii) in China: Conservation implications. Glob. Ecol. Conserv. 2021, 31, e01817. [CrossRef]

Lines: 567-568

29. Duan, X.G.; Li, J.Q.; Wu, S.H. MaxEnt Modeling to Estimate the Impact of Climate Factors on Distribution of Pinus densiflora. Forests. 2022, 13, 402. [CrossRef]

Lines: 604-605

“47. Fang, B.; Zhao, Q.; Qin, Q.L.; Yu, J. Prediction of Potentially Suitable Distribution Areas for Prunus tomentosa in China Based on an Optimized MaxEnt Model. Forests. 2022, 13, 381. [CrossRef]”

Comment 6: Though authors adopted a good way to reduce the number of variables, but look at Fig. 3A, many variables (whose sea green bars are matchable with the red Bar) depicting that their removal will not remarkably affect the training gain, and at the moments, their inclusion is contributing to enhance the model complexity than the model training gain.

Response: Thank you for your comments. The addition of variables will increase the complexity of the model, but the variables with high correlation degree will lead to the decrease of the validity of the model, which will affect the accuracy of the prediction results of the suitable area of Ginkgo biloba L., resulting in too large error. Therefore, we screened according to the correlation between variables (corresponding to lines 167-170 of the revised file). Specific as follows.

Lines: 167-170

Secondly, for the remaining environment variables, Spearman correlation analysis was performed in SPSS (IBM Statistics 26). Then, environmental variables with a correlation0.8 were selected. For the correlation above 0.8, the large contribution was retained [45]

Comment 7: AUC and TSS are indeed two widely used metrics as shown in literature, but have several inherent flaws, especially for Gaussian point processes, such as MaxEnt. Evaluate your models by using other metrics, such as partial AUC-ROC and AUC ratios etc. AUC and TSS are frequently criticized by the researchers (search for; 1. AUC: a misleading measure of the predictive distribution models; 2. Without quality presence-absence data, discrimination metrics such as TSS can be misleading measures of model performance) due to their dependence on prevalence.

Response: Sincerely thank you for your comments. The AUC evaluation index can well reflect the performance of the model without the influence of sample distribution and classifier bias. AUC inherits the excellent characteristics of the ROC curve evaluation index without manually setting the threshold. It is independent of threshold selection and has good interpretability. It can directly measure the performance of the model as a whole. The details are as follows. Lin et al.evaluated the model performance by the AUC value of the MaxEnt model and predicted the impact of future climate change on the distribution of Bandicota indica [54]. Mahmoodi et al. used the area under the curve (AUC) of the receiver operating characteristic (ROC) curve analysis to evaluate the effectiveness of the MaxEnt model, proving that the MaxEnt model can effectively predict the distribution of European yew in the study area [31]. In view of the fact that many scholars have proved that AUC can effectively evaluate the performance of the MaxEnt model, we use AUC to evaluate the model in this paper (corresponding to lines 189-194, 562-564, 567-568, 571-573 and 610-619 of the revised file). The details are as follows:

Lines: 189-194

The ROC evaluates the accuracy of the model with the area under the curve (AUC). ROC analysis method has been widely used in the evaluation of species distribution prediction models [27,29,31,50-51]. AUC value of 0.5-0.6 was predicted failure; 0.6-0.7 indicates poor prediction; 0.7-0.8 represents a general forecast; 0.8-0.9 indicates good prediction; 0.9-1.0 represents prediction results are very accurate and reliable [52]. When AUC > 0.85, the predicted results are acceptable [53-54].

Lines: 562-564

27. Gebrewahid, Y.; Abrehe, S.; Meresa, E.; Eyasu, G.; Abay, K.; Gebreab, G.; Kidanemariam, K.; Adissu, G.; Abreha, G.; Darcha, G. Current and future predicting potential areas of Oxytenanthera abyssinica (A. Richard) using MaxEnt model under climate change in Northern Ethiopia. Ecol. Process. 2020, 9, 6. [CrossRef]

Lines: 567-568

29. Duan, X.G.; Li, J.Q.; Wu, S.H. MaxEnt Modeling to Estimate the Impact of Climate Factors on Distribution of Pinus densiflora. Forests. 2022, 13, 402. [CrossRef]

Lines: 571-573

31. Mahmoodi, S.; Ahmadi, K.; Heydari, M.; Karami, O.; Esmailzadeh, O.; Heung, B. Elevational shift of endangered European yew under climate change in Hyrcanian mountain forests: Rethinking conservation-restoration strategies and management. Forest Ecol. Manag. 2023, 529, 120693. [CrossRef]

Lines: 610-619

“50. Wang, Y.S.; Xie, B.Y.; Wan, F.H.; Xiao, Q.M.; Dai, L.Y. Application of ROC curve analysis in evaluating the performance of alien species potential distribution models. Biod. Sci. 2007, 15, 365-372. [CrossRef]

  1. Şen, İ.; Sarıkaya, O.; Örücü, Ö.K. Predicting the future distributions of Calomicrus apicalis Demaison, 1891 (Coleoptera: Chrysomelidae) under climate change. J. Plant Dis. Protect. 2022, 129, 325-337. [CrossRef]
  2. Araújo, M.; Pearson, R.; Thuiller, W.; Erhard, M. Validation of species-climate impact models under climate change. Glob. Change Biol. 2005, 11, 1504-1513. [CrossRef]
  3. Mapunda, K.K., Andrew, S.M. Predicting the distribution of critically endangered tree species Karomia gigas under climate change in Tanzania. Ecol. Eng. 2023. 195, 107065. [CrossRef]
  4. Lin, S., Yao, D., Jiang, H., Qin, J., Feng, Z. Predicting current and future potential distributions of the greater bandicoot rat (Bandicota indica) under climate change conditions. Pest Manag Sci. 2023. 7804. [CrossRef]

Comment 8: As the targeted species is endemic to China, it is recommended that authors should draw its EOO based on the alpha-hull method using the available occurrences, as suggested by IUCN. This can be done by using ConR in R.

Response: Sincerely thank you for your comments. The results of the research and analysis in this paper include the spatial distribution of Ginkgo biloba L. in China under the current climate scenario and the possible spatial distribution of Ginkgo biloba L. in China under the future climate scenario. The distribution of Ginkgo biloba L. in the current and future periods is obtained by MaxEnt simulation, which is consistent before and after. It is convenient and fast to divide the suitable area and carry out superposition analysis. Although the calculation results of EOO index can show the distribution of species, the results in this paper cannot be superimposed with the possible spatial distribution of Ginkgo biloba L. in the future. In addition, the new version R does not include the ConR module (https://cran.r-project.org/web/packages/ConR/index.html), the latest update of the EOO indicator on the IUCN also remained in December 2019 (https://www.biodiversitya-z.org/content/species-area-of-distribution), is more difficult to implement. The results output from the MaxEnt model in this paper are feasible for the realization of the research objectives, and can reflect the current and future distribution changes of Ginkgo biloba under the influence of climate.

Comment 9: As you targeted the terrestrial biota, why NDVI or related measures were not considered to be used as important discriminatory variables? Such measures are very important for species residing in dense forests.

Response: Sincerely thank you for your comments. After considering various factors affecting plant growth, we refer to the relevant literature to consider the availability of variable data for the current and future periods. We chose topography, climate and soil to model the distribution of Ginkgo biloba L. These selected environmental variables can be obtained by sharing data, and the factors considered are relatively comprehensive, so we selected only these three aspects for analysis at present. NDVI is an important indicator, but the future NDVI data is not easy to predict, so we currently select climate, soil and topography variables for analysis (corresponding to lines 128-132 and 585-586 of the revised file). The details are as follows.

Lines: 128-132

“Plant growth is closely related to climate, topography, soil properties, vegetation types, NDVI and so on. In this study, the current and future distribution patterns of Ginkgo biloba L. will be simulated. Considering the availability of data in these periods (such as NDVI data in the future period cannot be obtained), we selected three environmental variables: bioclimate, soil and topography [38].”

Lines: 585-586

“38. Zhao, H.R, Yang, X.T., Shi, S.Y., Xu, Y.D., Yu, X.P., Ye, X.P. Climate-driven distribution changes for Bashania fargesii in the Qinling Mountains and its implication for panda conservation. Glob. Ecol. Conserv. 2023. 46, e02610. [CrossRef]”

Comment 10: To minimize the impact of sampling bias, a bias file (using species occurrence data and environment to estimate and develop a two-dimensional kernel density raster) for each species can be generated and used in MaxEnt distribution modelling. The inclusion of such bias files in the MaxEnt modelling effectively manipulate the background, and introduce the same spatial bias like that which exists in the presence data.

Response: Sincerely thank you for your comments. Your suggestion is very enlightening and constructive for our research in this area, we will carefully consider your suggestions in our future research. However, in this paper, we refer to the data, the wild Ginkgo biloba L. is distributed in the temperate and subtropical climate zones of China, and the distribution boundary is “north to Shenyang, Liaoning Province, south to Guangzhou, Guangdong Province, southeast to Nantou, Taiwan Province, west to Changdu, Tibet Autonomous Region, east to Putuo Island, Zhoushan, Zhejiang Province”, this is consistent with the distribution range of sample point data. The distribution of sample data in this paper is relatively balanced and is less affected by sampling bias. For the limitations of the MaxEnt model in this regard, we explained in the discussion (corresponding to lines 412-414 and 644-645 of the revised file). The details are as follows.

Lines: 412-414

The sampling deviation of the model background points caused by the obvious ten-dency of the sample distribution will also affect the prediction results [66].”

Lines: 644-645

“66. Alatawi, A.S., Gilbert, F., Reader, T. Modelling terrestrial reptile species richness, distributions and habitat suitability in Saudi Arabia. Journal of Arid Environments. 2020. 178, 104153. [CrossRef]”

Comment 11: The authors need to revise the text considering as they are communicating prediction values and not the sure values. e.g. “The area will increase by” should be revised to “potential land-area might increase to” throughout the text.

Response: Sincerely thank you for your comments. According to your suggestion, we express the possibility of the predicted value in the full text. The modification part mainly focuses on section 3.3.2 and 3.4, it is not possible to list them one by one. We list some here, you can see all in the revised file.

Lines: 293-297

Under different climate change scenarios, Ginkgo biloba L. suitable area may expand compared to the current and is relatively stable. By the future 2050s period, the total Ginkgo biloba L. suitable area may increase by 27.94%, 33.42%, 31.19% and 38.40% under the four climate scenarios, compared to the current one.”

Lines: 327-329

By the future 2070s period, the expansion of the Ginkgo biloba L. suitable area maybe 55.44×104 km2, 66.20×104 km2, 120.17×104 km2 and 102.76×104 km2 under different climate scenarios, respectively.”

Lines: 349-353

Under the ssp585 scenario, the center of gravity of Ginkgo biloba L. suitable area in the 2050s may move approximately to the boundary of Hubei Province, Shanxi Province and Chongqing City (109.7°E, 31.8°N), and the center of gravity of Ginkgo biloba L. suitable area in the 2070s may move to the southwestern part of Shanxi Province (107.4°E, 32.7°N).

Comment 12: Please revise Bio6, Bio13 to their full names in text.

Response: Sincerely thank you for your comments. The full names of the 19 bioclimatic variables mentioned in this paper are attached to the supplementary file. We also explain the full names of the three main climatic variables (Bio6, Bio13, Bio4) that affect the changes of the suitable areas of Ginkgo biloba L. (corresponding to lines 224-226 of the revised file). The details are as follows.

Lines: 224-226

The output results of the model (Table 1) showed that the contribution of each environmental variable 3% are Bio6 (Min Temperature of Coldest Month, 43.5%), Bio13 (Precipitation of Wettest Month, 21.8%), DEM (4.6%), and Bio4 (Temperature Seasonality, 3%),

Comment 13: Similarly, discussion section is a bit weak, and do not cover the targeted study objectives and research questions appropriately. Please cite the updated and last 5 year literature.

Response: Thank you for your comments. Based on your proposal, we refer to the relevant literature to enrich the discussion section and quote the latest literature in the past 5 years (corresponding to lines 371-375、385-392、430-434、627-630、637-647and 651-657 of the revised file). The details are as follows.

Line: 371-375

It is worth noting that Ginkgo biloba L. is stable in southeastern China, including eastern Sichuan, Chongqing, Hubei, Hunan, Jiangxi, Zhejiang, and northern Fujian, because the complex topography of these regions is prone to regional microclimates, which can mitigate climate change in the region and create stable climatic conditions for plants [59].

Line: 385-392

Under climate change, the predicted potential suitable area of Ginkgo biloba L. moved northward obviously, indicating that the northern boundary of subtropical and temperate zones in China may move northward. Some areas in the north have formed an environment suitable for the growth of Ginkgo biloba L. due to the changes in temperature. The topography of the Inner Mongolia Plateau and the Greater Khingan Range limits the growth of Ginkgo biloba L., so the predicted potential suitable areas for Ginkgo biloba L. are distributed in their south.

Line: 430-434

The potential geographical distribution pattern of Ginkgo biloba L. predicted by the simulation can be used as the base map for the protection planning of Ginkgo biloba L. The main environmental variables affecting the growth of Ginkgo biloba L. obtained from the research results provide a reference for artificial cultivation of Ginkgo biloba L.

Line: 627-630

59. Sanczuk, P., De, P.K., De, L.E., Luoto, M., Meeussen, C., Govaert, S., Vanneste, T., Depauw, L., Brunet, J., Cousins, S.A.O., Gasperini, C., Hedwall, P., Iacopetti, G., Lenoir, J., Plue, J., Selvi, F., Spicher, F., Uria-Diez, J., Verheyen, K., Vangansbeke, P., De, F.P. Microclimate and forest density drive plant population dynamics under climate change. Nat. Clim. Chang. 2023, 13(8), 840-847. [CrossRef]

Line: 637-647

“63. Khan, A.M., Li, Q.T, Saqib, Z., Khan, N., Habib, T., Khalid, N., Majeed, M., Tariq, A. MaxEnt Modelling and Impact of Climate Change on Habitat Suitability Variations of Economically Important Chilgoza Pine (Pinus gerardiana Wall.) in South Asia. Forests. 2022, 13(5). [CrossRef]

  1. Amaro, G., Fidelis, E.G., da, S.R.S., Marchioro, C.A. Effect of study area extent on the potential distribution of Species: A case study with models for Raoiella indica Hirst (Acari: Tenuipalpidae). Ecol. Modell. 2023, 483, 110454. [CrossRef]
  2. Gong, H.D., Cheng, Q.P., Jin, H.Y., Ren, Y.T. Effects of temporal, spatial, and elevational variation in bioclimatic indices on the NDVI of different vegetation types in Southwest China. Ecol. Indic. 2023. 154, 110499. [CrossRef]
  3. Alatawi, A.S., Gilbert, F., Reader, T. Modelling terrestrial reptile species richness, distributions and habitat suitability in Saudi Arabia. J. Arid Environ.. 2020. 178, 104153. [CrossRef]
  4. Zahoor, B., Liu, X., Songer, M. The impact of climate change on three indicator Galliformes species in the northern highlands of Pakistan. Environ Sci Pollut Res Int. 2022. 29(36), 54330-54347. [CrossRef]”

Line: 651-657

69. Xu, W.H., Jiang, J., Lin, H.Y., Chen, T.Y., Zhang, S.Y., Wang, T.L. Assessment of the impact of climate change on endangered conifer tree species by considering climate and soil dual suitability and interspecific competition. Sci. Total Environ. 2023. 877, 162722. [CrossRef]

  1. Mihai, G., Alexandru, A.M., Nita, I.A., Birsan, M.V. Climate Change in the Provenance Regions of Romania over the Last 70 Years: Implications for Forest Management. Forests. 2022. 13(8), 1203. [CrossRef]
  2. Nelson, K.N., ODean, E., Knapp, E.E., Parker, A.J., Bisbing, S.M. Persistent yet vulnerable: resurvey of an Abies ecotone reveals few differences but vulnerability to climate change. Ecology. 2021. 102, 12, e03525. [CrossRef]

Round 2

Reviewer 4 Report

Comments and Suggestions for Authors

The authors have performed all the suggested revisions appropriately, and accordingly, I recommend this MS for publication.

Regards

Comments on the Quality of English Language

Minor editing of the English language is required.

Author Response

Thank you for your comments.